# Naturally ornate RNA-only complexes revealed by cryo-EM

Rachael C. Kretsch[1], Yuan Wu[2], Svetlana A. Shabalina[3], Hyunbin Lee[4], Grace Nye[5], Eugene V. Koonin[3], Alex Gao[1,4], Wah Chiu[1,5,6,7 ✉] & Rhiju Das[1,2,4 ✉]

The structures of natural RNAs remain poorly characterized and may hold numerous surprises[1–4]. Here we report three-dimensional structures of three large ornate bacterial RNAs using cryo-electron microscopy (cryo-EM). GOLLD (Giant, Ornate, Lake- and Lactobacillales-Derived), ROOL (Rumen-Originating, Ornate, Large) and OLE (Ornate Large Extremophilic) RNAs form homo-oligomeric complexes whose stoichiometries are retained at lower concentrations than measured in cells. OLE RNA forms a dimeric complex with long co-axial pipes spanning two monomers. Both GOLLD and ROOL form distinct RNA-only multimeric nanocages with diameters larger than the ribosome, each empty except for a disordered loop. Extensive intramolecular and intermolecular A-minor interactions, kissing loops, an unusual A–A helix and other interactions stabilize the three complexes. Sequence covariation analysis of these large RNAs reveals evolutionary conservation of intermolecular interactions, supporting the biological importance of large, ornate RNA quaternary structures that can assemble without any involvement of proteins.

The importance of non-coding RNAs (ncRNAs) across biology is increasingly recognized, but only a small number have been functionally characterized, with studies revealing sophisticated catalytic and sensory functions in some cases[5–9]. Bacteria, archaea and their viruses are thought to possess numerous diverse and complex ncRNAs, but most of these have not been thoroughly studied[1–4]. Furthermore, there is a conspicuous shortage of data on the 3D structures of RNA molecules. Out of more than 4,000 RNA classes in the RNA Families (RFAM) database 15.0, only 143 have experimentally resolved tertiary structures[10]. For many of the remaining cases, it appears likely that structural characterization will depend on reconstitution of the RNA with small molecule, protein or nucleic acid partners, which are unknown in most cases.

The Breaker laboratory and collaborators have previously described three classes of bacterial and phage RNAs for which covariance analysis of genomic and metagenomic sequences revealed secondary structures that were so extensive and elaborate that 'ornate', 'giant' or 'large' were included in their names: GOLLD RNA[3], ROOL RNA[1,11] (concomitantly reported in ref. 12) and OLE RNA[2]. The functions of these three classes of large RNAs remain poorly understood.

Here, using cryo-EM, we show that OLE, ROOL and GOLLD all form atomically ordered 3D structures. Unexpectedly, the three structures are stabilized not by proteins but by other copies of the same RNA molecule in ornate quaternary assemblies with many intermolecular bridges, a phenomenon that has not previously been observed for natural RNA molecules[13].

## OLE forms an RNA-only dimer

OLE is a class of large RNAs with an ornate secondary structure that is conserved throughout evolution[2]. OLE is found mainly in extremophilic bacteria, and experimental characterization in *Halalkalibacterium halodurans* has demonstrated its involvement in integrating energy availability, metal ion homeostasis and drug treatment to mediate cellular adaptation, although the underlying molecular mechanisms remain unknown[2,14–16]. Cellular localization to the membrane, binding to at least six protein partners[15,17–22] and evidence of alternative secondary structures[17] suggested that OLE was unlikely to form a well-defined RNA-only 3D structure. However, our study showed that the 577-nucleotide (nt) OLE RNA from *Clostridium acetobutylicum*[2,23] formed distinct, compact particles that were clearly visible in cryo-EM images (Fig. 1a). Furthermore, a 2.9 Å resolution 3D map of a dimeric OLE RNA could be reconstructed with two-fold imposed symmetry (Extended Data Fig. 1). A model of the each chain has been built for 308 nt in the OLE 5' region, with Q-scores[24] exceeding the expected score at this resolution (Fig. 1b, Extended Data Fig. 1e and Supplementary Video 1).

Our OLE dimer map shows that it is organized as a series of parallel A-form helices, resembling a bundle of pipes. The exterior ends of these pipes from each chain are interconnected into a five-way junction, with a secondary structure that agrees with the previously proposed one for the observed domain with stems P3 to P9.3 (refs. 2,15) (Fig. 1c; hereafter, paired stems, hairpin loops and joining linkers are designated 'P', 'L' and 'J', respectively, following conventional RNA nomenclature).

[1]Biophysics Program, Stanford University, Stanford, CA, USA. [2]Howard Hughes Medical Institute, Stanford University, Stanford, CA, USA. [3]Computational Biology Branch, Division of Intramural Research, National Library of Medicine, National Institutes of Health, Bethesda, MD, USA. [4]Department of Biochemistry, Stanford University School of Medicine, Stanford, CA, USA. [5]Division of CryoEM and Bioimaging, SSRL–SLAC National Accelerator Laboratory, Menlo Park, CA, USA. [6]Department of Bioengineering and James Clark Center, Stanford University, Stanford, CA, USA. [7]Department of Microbiology and Immunology, Stanford University, Stanford, CA, USA. ✉e-mail: wahc@stanford.edu; rhiju@stanford.edu

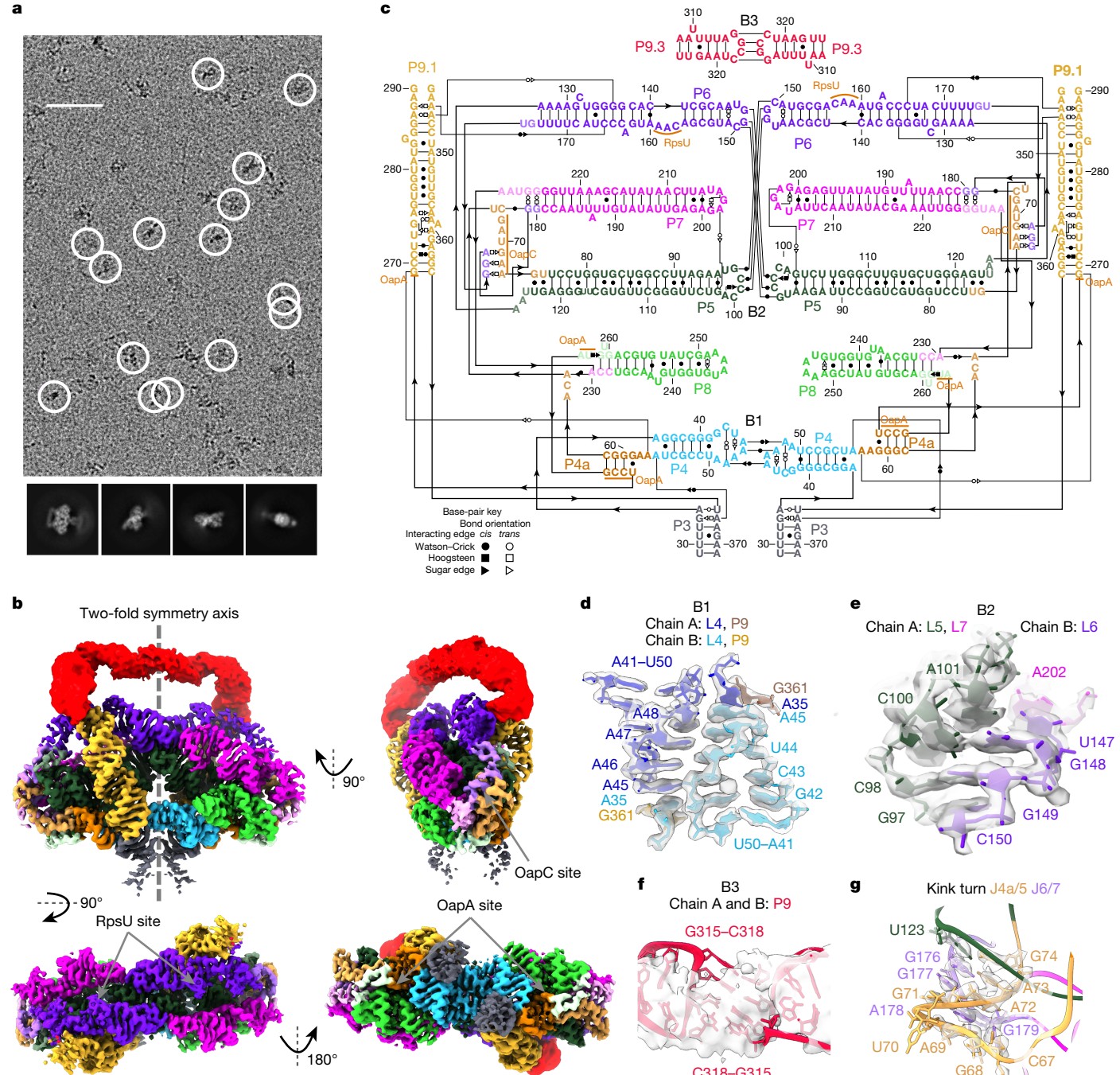

**Fig. 1 | Structure of OLE homodimer. a**, Top, representative micrograph (6,752 micrographs total). Particles selected for reconstruction are circled in white. Bottom, 2D class averages. Scale bar, 50 nm. **b**, The cryo-EM reconstruction of the OLE dimer. The top left image depicts the separation of the two chains: one chain on the right and the other on the left. Each domain of both chains is coloured. To aid visualization, the flexible P9.3 domain (red) is displayed with the unsharpened map at 10σ contour. The proposed binding sites of previously described proteins (RpsU, OapC and OapA) are labelled. **c**, Secondary structure of OLE dimer. The domains are coloured as in **b**. **d**–**f**, The intermolecular bridge interactions B1 (**d**), B2 (**e**) and B3 (**f**), coloured by domain. In **d**, the domain colouring is darker for chain A to differentiate the chains. **g**, The kink-turn motif that may bind the OapC protein, identical for each monomer. The sharpened cryo-EM map is displayed at the following contours: 7σ (**b**), 15σ (**d**), 12σ (**f**) and 10σ (**g**).

An unusual but highly conserved symmetric interaction comprised of four A–A base pairs between two chains (L4, Fig. 1d), intermolecular base pairing and stacking interactions connecting L5, L6 and L7 (Fig. 1e), and a kissing loop (L9.3, Fig. 1f) 'weld' the pipes together in the middle of the complex. We denote these intermolecular interactions 'bridges' B1–B3, as used in ribosome nomenclature[25]. An elaborative list of intramolecular motifs and intermolecular interactions is presented in Supplementary Tables 1 and 2, respectively. Beyond the 5' region, other conserved parts of OLE were not resolved in the structure, suggesting flexibility.

Surprisingly, regions of OLE that were previously thought to adopt alternative structures upon protein binding are clearly resolved and solvent-accessible, suggesting that proteins may bind the OLE dimer in the pre-formed RNA conformation that we observed here. Our cryo-EM data show that these proteins are not required for the folding of the 5' domain of OLE, and the RNA structure itself may have a crucial role in

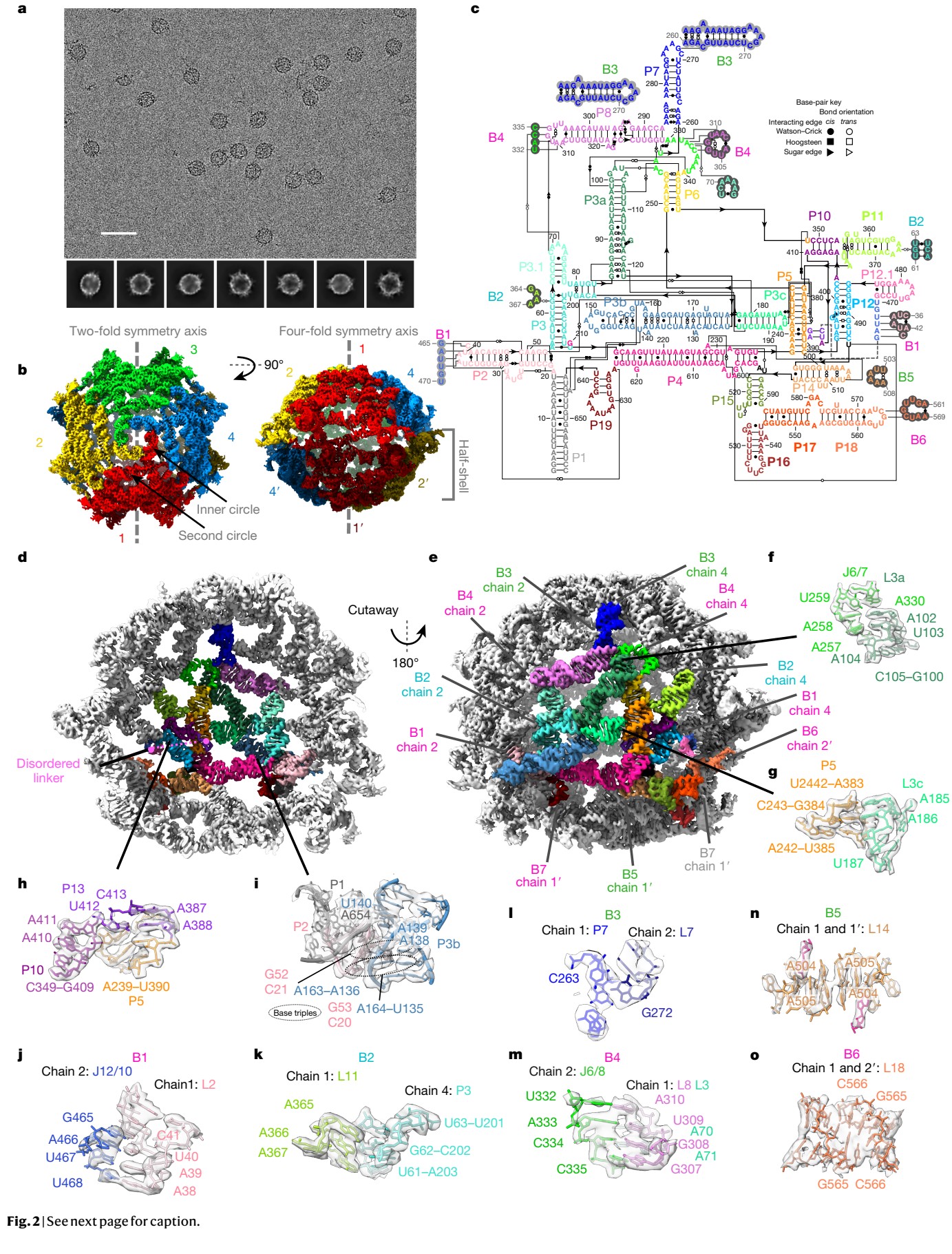

**Fig. 2** | See next page for caption.

**Fig. 2 | Atomically ordered structure of ROOL homo-octamer.**
**a**, Representative micrograph (top; 4,462 micrographs total) with 2D class averages (bottom). Scale bar, 50 nm. **b**, The 3.1 Å cryo-EM reconstruction of the ROOL complex with $D_4$ symmetry. The map is coloured by 8 labelled chains. In the top view (left), the inner and second circle are labelled. **c**, Secondary structure of ROOL coloured by domain. Only one chain is shown, in full. Nucleotides involved in intermolecular interactions have been circled in light or dark grey. **d,e**, Chain 1 is coloured by domain with all other chains in grey. **d**, Cutaway view, showing chain 1 from the interior of the nanocage with the disordered linker labelled in pink (nucleotides 414–464). **e**, Intermolecular interactions or bridges of chain 1 are labelled, with kissing loops labelled in magenta, A-minor interactions in cyan and other interactions in lime. One interaction, B7 (grey), is not ordered in this cryo-EM map, but residues come in sufficiently close contact that interactions could form. The same interactions, but between different pairs of chains, share the same number. **f–i**, Selected intramolecular interactions as labelled in **d,e. j–o**, Intermolecular interactions as labelled in **e**. The sharpened cryo-EM map is displayed at the following contours: $6\sigma$ (**b,d,e**), $8\sigma$ (**o**), $16\sigma$ (**g,i–n**) and $20\sigma$ (**f,h**).

organizing these proteins. In particular, the protein OapC was previously hypothesized to bind a kink turn between J4a/5 and J5/6, and binding of OapC was thought to alter secondary structure, in particular increasing protection of J6/7 to in-line hydrolysis[17]. Our OLE dimer structure supports formation of a kink turn[26,27] in J4a/5 at the base of P5, but this kink turn is formed with J6/7, not J5/6 (Fig. 1g). The previously observed protection of J6/7 may therefore be explained by direct binding to the protein, and not by a rearrangement of secondary structure. In addition, whereas the internal loop of the P6 stem is different from the previously proposed one, it exposes residues 163–165, which were proposed to bind the protein RpsU[15]. A163 is flipped out of the helix and docks into a pocket created by P5, P6, P7 and dimer interface. This OLE dimer pocket is reminiscent of the pocket RpsU occupies in the ribosome, supporting the previous hypothesis that OLE could sequester RpsU[15].

## ROOL assembles into an ordered nanocage

ROOL is a class of RNAs that is encoded in a wide variety of bacterial prophages and phages, often near tRNA islands[1,11,12]. The predicted secondary structure is highly complex with multiple pseudoknots, but no protein binding partners have been identified, leading to the hypothesis that ROOL may function as an RNA-only complex[1]. Although no function has been described for ROOL, it has been shown to be as abundant as 16S ribosomal RNA, but non-essential, in at least one strain of *Ligilactobacillus salivarius*[12].

The 659-nt ROOL env-120, discovered in cow rumen[1,28], produces visually clear, symmetric particles in cryo-EM micrographs (Fig. 2a and Extended Data Fig. 2). The 3.1 Å reconstructed map reveals a closed, hollow nanocage structure that comprises 8 chains with dihedral symmetry and a diameter of approximately 280 Å, larger than the maximal dimension of a bacterial ribosome (approximately 250 Å) (Fig. 2b and Supplementary Video 2). Each chain has a secondary structure that is consistent with the stems P1 to P19 proposed previously by covariation analysis[12], including the pseudoknot P10 (Fig. 2c). Atomic models for each chain can be built with a good match to the map density as shown by the Q-scores[24] (Extended Data Fig. 2e). Our model shows intramolecular tertiary interactions (Fig. 2d–i), which scaffold the flat monomer structure (Fig. 2f–i), including a set of non-canonical base pairs and stacking interactions that connect loops L3a and J6/7 (Fig. 2f), an A-minor interaction between L3c and P5 (Fig. 2g), an additional pseudoknot P13 (adjacent to P5 and P10, Fig. 2h), a complex set of non-canonical pairs between nucleotides that are already in stems P1, P2 and P3b (Fig. 2i), and other motifs (Supplementary Table 1).

The ROOL quaternary complex is an octameric nanocage, with a top and bottom half-shells, each formed by 4 chains, hereafter labelled chains 1–4 and 1′–4′. Within a half-shell, each chain forms 8 bridges with its neighbours, 4 on each side, labelled B1–B4 (Fig. 2j–o). Starting from the top, the loop of stem P7 forms an isolated base pair with a bulged out base in stem 7 of the next chain (B3, Fig. 2l). This 'daisy chain' of interacting stem-loops forms an inner circle on the top of the half-shell approximately 36 Å in diameter (Fig. 2b). The P7 stem is not always conserved, but a second circle of RNA (Fig. 2b), involving a quaternary kissing loop (B4, Fig. 2m), is highly conserved in evolution, and was identified as tertiary interaction by previous covariation analysis[1]. An A-minor interaction (B2, Fig. 2k) and a novel quaternary kissing loop (B1, Fig. 2j) further glue together the chains in the half-shell. Between a novel intramolecular tertiary interaction (Fig. 2g) and the intermolecular kissing loop B1 (Fig. 2j), we identified a disordered region that appears to be located inside the nanocage, based on the position of flanking regions (Fig. 2d and Extended Data Fig. 2d,e). This region was previously identified as a linker with little to no sequence or structural similarity across homologues[1].

As opposed to a simple dimer such as the OLE interface, where each chain interacts with a single partner, in the ROOL complex, each chain reaches over and interacts with two chains in the other half-shell. These interactions favour the full cage assembly, as opposed to isolated dimers. B5 and B6 are quaternary interactions in which the same sequences from different chains interact via adenosine stacking and Watson–Crick–Franklin base pairing, respectively (Fig. 2n,o). An additional interaction between the internal loop J17/18 of chain 1, previously proposed to form a pseudoknot with the flank of the linker region, and P19 of chain 1′ seems plausible given their proximity, but that region was not well-resolved in our structure.

## GOLLD assembles into a distinct nanocage

GOLLD RNAs are the largest among the three RNA classes analysed here, with many members exceeding 800 nucleotides in length[3,11,29]. GOLLD, similar to ROOL, is a molecule of unknown function encoded in bacterial prophages and phages, often near tRNA islands, but with sequences and secondary structures that are distinct from those of ROOL[3,11,29]. GOLLD expression has been shown to increase during the lysis of bacterial cells infected by phage[11]. Unlike ROOL, the predicted secondary structures of GOLLD RNAs consist of a universally conserved 3′ region and a less conserved 5′ region[3].

The GOLLD env-38 RNA, first identified in a marine metagenomic sample downstream of Met-tRNA[3,30], produces visually striking flower-like particles in cryo-EM micrographs (Fig. 3a and Extended Data Fig. 3). The 3D reconstruction at 3.0 Å resolution shows that GOLLD forms a nanocage, similar to the one formed by ROOL, but larger. The GOLLD structure is a closed 14-mer with $D_7$ quaternary symmetry, with a diameter of 380 Å and a completely empty interior except for a disordered loop (Fig. 3b,c and Supplementary Video 3). Models for each of the 14 chains were built with Q-score[24] in accordance with the map resolution (Extended Data Fig. 3g). As with ROOL, kissing loops and A-minor interactions underlie the tertiary and quaternary structure of GOLLD in addition to other motifs (Fig. 3d–t and Supplementary Tables 1 and 2) but the specific interactions are distinct. Beyond confirming the accuracy of the previously predicted secondary structure with stems P1–P27 (Fig. 3d), the tertiary structure of GOLLD reveals prominent interactions, including A-minor interactions involving adenosines at the P3–P4–P5 junction (Fig. 3g), an A-minor interaction between adenosines in L26 and stem P14 (Fig. 3h) and a loop L22 that forms a pseudoknot with the nearby linker J17/22, in addition to an A-minor interaction with that pseudoknot (Fig. 3i). Furthermore, loop L27 brings together seven regions by forming base pairs with stem P23 and linker J24/26 as well as base-backbone interactions with

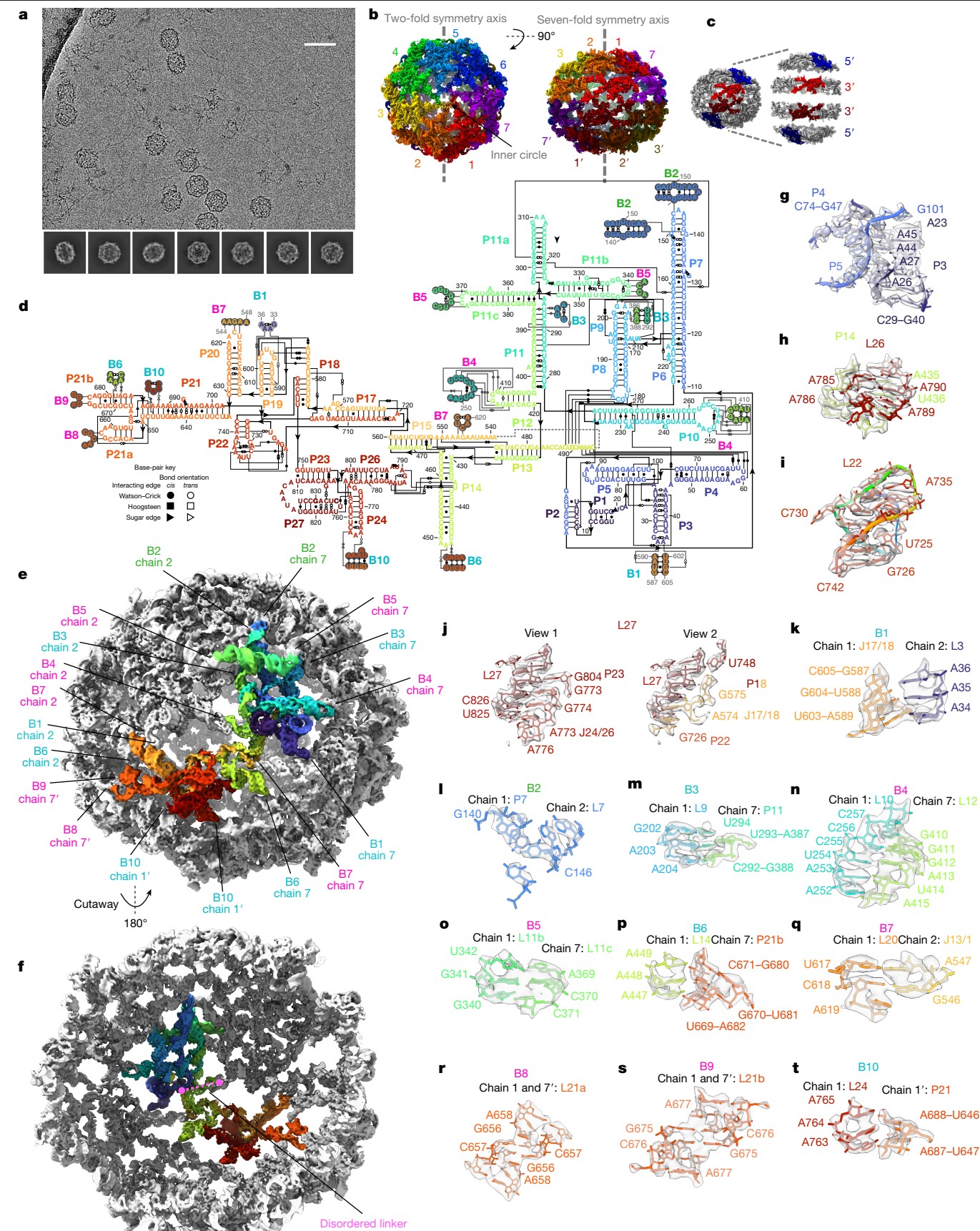

**Fig. 3** | See next page for caption.

**Fig. 3 | Atomically ordered structure of GOLLD homo-14-mer.**
**a**, Representative micrograph (23,281 micrographs total) and 2D class averages. Scale bar, 50 nm. **b**, The 3.0 Å cryo-EM reconstruction of GOLLD with $D_7$ symmetry, coloured by chain. In the top view (left), the inner circle is labelled. **c**, The 5′ (blue, residues 1–420) and 3′ (red, residues 421–833) regions of GOLLD organize into the cap and a ring of the half-shell of the nanocage, respectively. To demonstrate the separation of domains, the four regions are artificially moved apart. **d**, The secondary structure of GOLLD. Only one chain is displayed in full, nucleotides participating in intermolecular interactions that are from other chains are circled in light or dark grey. **e**, One chain of GOLLD is depicted in rainbow, intermolecular interactions or bridges are labelled, with kissing loops labelled in magenta, A-minor interactions in cyan and other interactions in lime. The same interaction, but between different pairs of chains, share the same number. Each chain interacts with four other chains. **f**, Same as **e**, but rotated and is cut away to show the rainbow-labelled chain from the interior of the nanocage with the disordered linker labelled in pink (nucleotides 497–538). **g**–**j**, Select intramolecular interactions. **k**–**t**, Intermolecular interactions as labelled in **e**. The sharpened cryo-EM map is displayed at the following contours: $12\sigma$ (**e**,**f**,**i**), $14\sigma$ (**c**,**o**), $16\sigma$ (**b**,**n**), $18\sigma$ (**s**,**q**), $20\sigma$ (**k**–**m**,**p**), $22\sigma$ (**t**), $25\sigma$ (**h**,**i**) and $30\sigma$ (**g**,**j**).

two additional stems, P18 and P22, and linker J17/18 (Fig. 3j). Similar to ROOL, the variable linker within each chain is not resolved, but the positions of immediate flanking sequences in the 5′ and 3′ regions indicate that the linker resides in the interior of the cage (Fig. 3f and Extended Data Fig. 3f,g). Globally, the cryo-EM structure shows that the 5′ region and the 3′ region form separate domains in the 3D structure (Fig. 3c). This separation could explain why the 3′ and 5′ domains are divergent in GOLLD, whereas, in ROOL, the 5′ and 3′ regions are intertwined and hence have to co-evolve to maintain the tertiary and quaternary structure.

The 5′ domains of GOLLD form the cap of each half-shell of the nanocage. Within the cap of each half-shell, each of 7 monomers forms 8 quaternary bridges to other chains—4 on each side, including kissing loops, A-minor interactions and other interactions (B2–B5, Fig. 3e,l–o). B2 (Fig. 3l) closely resembles the daisy chain of interacting stem-loops from ROOL, except that the distance between the pairs of interacting residues is reduced from 9 nt to 4 nt. This compensates for the increased number of chains in GOLLD, resulting in an inner circle of roughly the same diameter as of ROOL. In GOLLD, the only non-interacting loop with a conserved sequence, L11a (sequence GAAA), points towards this inner circle. The 3′ regions complete the half-shell below this 5′ cap through two interactions: an A-minor interaction (B6, Fig. 3p) and a kissing loop between L20 and J13/15, which was previously identified by covariation analysis[3] and here shown to be an intermolecular bridge (B7, Fig. 3q). Only a single intermolecular A-minor interaction, B1, glues the 3′ and 5′ regions from different chains together (Fig. 3k).

Finally, similar to the ROOL nanocage, the two half-shells come together with each chain in the top half-shell interacting with two chains in the bottom half-shell. In the GOLLD nanocage, these interactions consist of two self-interacting kissing-loop interactions (B8 and B9, Fig. 3r,s) and an A-minor interaction (B10, Fig. 3t) involving 3′ regions from different chains.

## Biological relevance of homo-multimers

Symmetric multimers are common among proteins and rationally designed RNA molecules[26,31–33], but observations of natural RNA multimers are rare. When observed, natural RNA homomeric interactions typically involve a single contact[13]. Further, with notable exceptions of viruses, such as HIV and other retroviruses[34] and the Φ29 bacteriophage[35–37], the biological relevance of RNA homomeric complexes has not been demonstrated, leaving the possibility that they form only at high RNA concentrations and extreme ionic conditions or in the context of the specific constructs chosen for in vitro structural characterization. By contrast, several lines of evidence support GOLLD, ROOL and OLE forming multimers in their biological contexts.

First, concomitant with the same set of cryo-EM studies presented above, we resolved a 2.9 Å resolution map of another large RNA molecule, the raiA motif from *C. acetobutylicum*[23,38], as a pure monomer (Extended Data Figs. 4 and 5 and Supplementary Text 1), refuting the possibility that any large RNA would form a multimer in our experimental conditions. Independent studies have also resolved the raiA motif as a monomer[39,40]. Additionally, we characterized the 343-nt HNH

endonuclease-associated RNA and open reading frame (HEARO)[39] from *Limnospira maxima*, which is known to form a defined RNA structure that is involved in DNA nickase activity when bound to the protein IsrB[41]. Unlike the 5′ region of OLE, which is also known to bind proteins, the HEARO RNA was disordered in the absence of the protein (Extended Data Fig. 6), suggesting that multimer formation of protein-binding RNAs is not an artefact of cryo-EM experimental conditions.

Second, mass photometry, which gives high precision estimates of molecular weight but requires molecular binding to surfaces, confirms the stoichiometry of GOLLD, ROOL and OLE to be 14, 8 and 2, respectively, at RNA concentrations as low as 12.5 nM (Extended Data Fig. 7). This concentration is three orders of magnitude lower than the concentrations in our cryo-EM experiments and corresponds to a population of only around ten RNA molecules in a bacterial cell, substantially lower than what is expected from observed expression levels[1].

Third, using dynamic light scattering (DLS; Extended Data Fig. 7), we confirmed that both ROOL and GOLLD primarily form thermostable multimers, with no detectable fraction of monomers, at temperatures up to 55 °C and concentrations as low as 110 nM.

Fourth, for all three structures each chain contains five or more conserved inter-subunit contacts, indicative of intricate arrangements that suggest selection pressure during the evolution of these RNAs.

Fifth, using comparative analysis of both sequences and secondary structures, we detected evolutionary conservation of structural elements and, in particular, the sites of intermolecular interactions supporting RNA homo-oligomerization (Supplementary Files 1–3 and Supplementary Table 3). Comparative analysis of OLE, ROOL and GOLLD showed that, although the sequences of these RNAs are not highly conserved, all intramolecular stems exhibit extensive base pairing supported by covariation analysis, including stems whose loops are involved in intermolecular bridges (Extended Data Fig. 8a–c, Supplementary Text 2 and Supplementary Table 3). The A positions in the OLE non-canonical A–A base-pair stem bridge B1 and the GOLLD A-minor interaction bridge B6 are highly conserved (Figs. 1d and 3p and Extended Data Fig. 8d,e). The intermolecular base pairs between ROOL J6/8 and L8 (bridge B4, Fig. 2m) were detected as a prominent, conserved quaternary interaction in prior covariation analysis[40] (Extended Data Fig. 8b and Supplementary Table 3). In other bridges, we observed intermolecular symmetric kissing loops that had base pairs between the same loop from two different chains: nucleotides 315–318 in chain A and B of OLE (B3) and nucleotides 656–657 from chain 1 and 7′ of GOLLD (B8). Apparent covariance at immediately adjacent nucleotides in these loop sequences supports intermolecular base pairs because base pairing of adjacent nucleotides within the same chain is stereochemically precluded (Extended Data Fig. 8f,g). OLE L9.3 and GOLLD L21a were each found to covary in this manner, switching an internal tetranucleotide between palindromes GGCC to GAUC or AGCU and an internal dinucleotide between GC and CG, supporting bridges OLE B3 and GOLLD B8, respectively (Extended Data Fig. 8f,g). The other symmetric kissing loops in our structures, GOLLD L21b (bridge B9) and ROOL L18 (bridge B6), were highly conserved across the variants for which the loops could be confidently aligned (Extended Data Fig. 8g,h), precluding similar

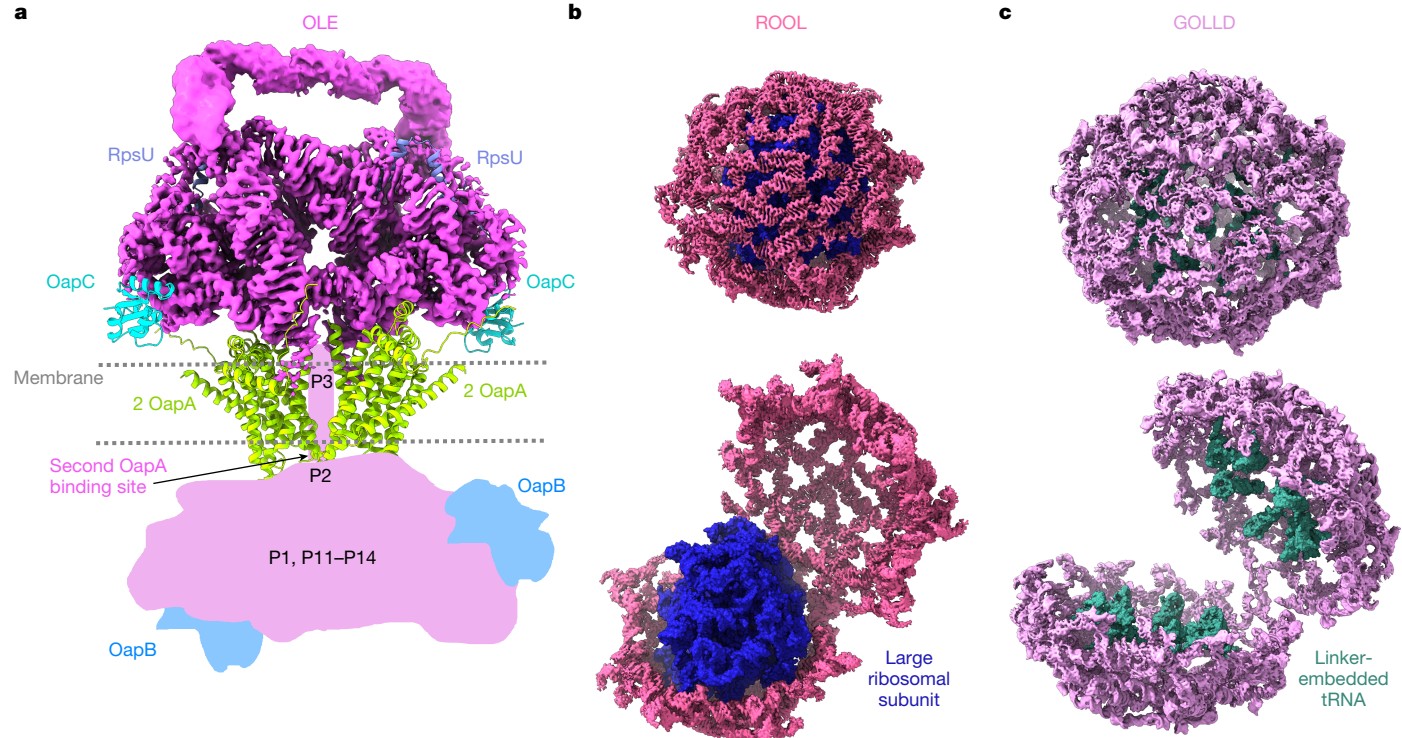

**a** OLE

RpsU · RpsU

OapC · OapC

Membrane

2 OapA · 2 OapA

Second OapA
binding site

P3

P2

P1, P11–P14

OapB

OapB

**b** ROOL

**c** GOLLD

Large
ribosomal
subunit

Linker-
embedded
tRNA

**Fig. 4 | Structure-guided hypotheses for homo-oligomeric RNAs. a**, OLE dimer displayed with AlphaFold 3[47] models of OapC, OapA dimer and RpsU proteins at their proposed binding sites. **b,c**, The two half-shells of the RNA nanocages are held together by only a few interactions and hence could open up to encapsulate other biomolecules, such as protein, metabolites, nucleic acids or RNA–protein complexes. **b**, The RNA nanocage formed by ROOL is shown encapsulating a ribosomal large subunit. **c**, The RNA nanocage formed by GOLLD is shown exposing the covalently linked tRNAs when open. The sharpened cryo-EM maps are displayed at the following contours 7σ (**a**), 6σ (**b**) and 12σ (**c**).

covariance analysis but consistent with the importance of the observed intermolecular interactions.

## Discussion

Together, our cryo-EM data, biophysical experiments and evolutionary analyses show that GOLLD, ROOL and OLE each form not only ornate secondary structures but also symmetric quaternary assemblies stabilized by many intermolecular contacts. While this Article was being revised, a publication appeared reporting similar cryo-EM structures, supporting the reproducibility of cryo-EM[42]. These structures and their complex network of RNA structure motifs offer a rich source of data for RNA structure prediction and design efforts. OLE forms a dimer shaped like a bundle of pipes and exposes structured binding pockets for protein partners such as the membrane-associated OLE-associated protein A (OapA). After superimposing an OapA dimer to each P4a site (OapA is known to bind OLE in a 2:1 ratio[41–43]), we note that the RNA could induce the formation of an OapA tetramer. OapA is a membrane protein, and the tetramer is reminiscent of the double-stranded RNA transporter SID-1[43–45], suggesting that it may be able to accommodate RNA elements, such as the 3′ region of OLE, which was not resolved here (Fig. 4a). In contrast to OLE, and despite unrelated sequences and distinct secondary and tertiary structures, GOLLD and ROOL both form nanocages, suggesting that their function might involve encapsulating their internal disordered linkers and/or other molecules, analogous to proteinaceous microcompartments that are common in bacteria and archaea[46]. Although not large enough to enclose entire DNA genomes of their parent phages, these cages might contain macromolecules of significant size (Fig. 4b,c), such as phage-encoded tRNAs, which are sometimes present in the GOLLD linker region, bacterial ribosomes, which have been shown to bind GOLLD in pull-down assays[45], metabolites, or stress response proteins. It remains to be determined whether nanocage formation is a common feature among large natural RNAs.

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

# Methods

## In vitro RNA synthesis

DNA templates containing the RNA sequence of interest prepended with the T7 promoter (see Supplementary Table 4 for sequences) were ordered as gBlocks from IDT. Primers designed to amplify these sequences (see Supplementary Table 4 for sequences) were also from IDT. PCR amplification was carried out with NEBNext Ultra II Q5 Master Mix (NEB M0544S) using 10 ng of template per reaction. The thermocycler settings were: 98 °C for 30 s, then 35 cycles of 98 °C for 10 s, 55 °C for 30 s, then 72 °C for 30 s, and a final step of 72 °C for 5 min. The PCR products were then column purified using the QIAquick PCR Purification Kit (Qiagen 28104) and run on a 2% E-Gel agarose gel (Thermo Scientific A42135) to check DNA quality. DNA concentration was measured using a NanoDrop. Purified DNA smaller than 515 bp were in vitro transcribed using TranscriptAid T7 High Yield Transcription Kit (Thermo Scientific K0441) with 6 µl of DNA template per reaction. Purified DNA longer than 515 bp were in vitro transcribed using MEGAscript T7 Transcription Kit (Thermo Scientific AM1334) with 6−8 µl DNA template per reaction. These in vitro transcription reactions were incubated at 37 °C for 6 h, then held at 4 °C before DNase treatment. The RNA was then purified using the RNA Clean & Concentrator-25 Kit (Zymo Research R1017) and eluted in 30 µl water. The concentration of purified RNA was measured using a NanoDrop, and the quality was checked using the Agilent 2100 Bioanalyzer (Nano RNA Assay, run by the Stanford PAN Facility; Bioanalyzer 2100 Expert B.02.11.SI824), as shown in Extended Data Fig. 7a.

## RNA folding

For all subsequent experiments, RNA was re-folded using the same basic protocol. RNA concentrations used and any other modifications to this standard protocol are mentioned in each section. RNA was denatured (90 °C for 3 min, room temperature for 10 min) in 50 mM Na-HEPES pH 8.0. RNA was then folded with 10 mM $MgCl_2$ at 50 °C for 20 min, and cooled to room temperature for at least 10 min before taking measurements.

## Mass photometry

Mass photometry data were collected using the Refeyn TwoMP, using AcquireMP version 2024-R1.1 and DiscoverMP version 2024-R1 to obtain histogram data. For the OLE data, coated glass slides from the MP Sample Preparation Pack (MP-CON-21014) were used, for the ROOL and GOLLD data Mass Glass UC slides (MP-CON-41001) were used after coating with poly-L-lysine. The instrument was focused using droplet-dilution. Data were collected for 1 min using the large image size. The contrast data were calibrated to nucleotide length using the Millennium RNA Markers (Thermo Scientific AM7150). Gaussians were fitted by the automatic analysis in DiscoverMP. The resulting data and plotting code can be found in the accompanying GitHub repository.

Mass photometry data were not reliable for the raiA motif. raiA motif RNA was folded at 1 µM following the standard procedure above. On the stage two dilutions were attempted, 15 µl buffer:2 µl sample for final concentration of 118 nM and 18 µl buffer:2 µl 10x diluted sample for a final concentration of 10 nM. There is a known issue with nucleic acid samples, whereby there are noisy low-mass peaks[48] (communication with the company Refeyn). These are not present in the buffer alone. For this reason, raiA motif (205 nt) is below the recommended minimal size for mass photometry, and indeed when we attempted to collect data on the raiA motif we observed noise peaks, containing the size of raiA motif monomer but smaller than any multimer, in both binding and unbinding regimes, indicating unreliable results (data not shown).

OLE was folded at 0.25 µM and was diluted to 12.5 nM on the stage. OLE folded in various buffers, all including 50 mM Na-HEPES pH 8.0, with other components added at the time when $MgCl_2$ is added in the standard protocol: (1) nothing added; (2) 1 mM $MgCl_2$; (3) 10 mM $MgCl_2$, standard; (4) 100 mM $MgCl_2$; (5) 10 mM $MgCl_2$ and 1% ethanol; (6) 10 mM $MgCl_2$ and 5% ethanol; (7) 0 mM $MgCl_2$ and 200 mM KCl; (8) 10 mM $MgCl_2$ and 200 mM KCl; (9) 0 mM $MgCl_2$ and 200 mM NaCl; and (10) 10 mM $MgCl_2$ and 200 mM NaCl. Buffers with $MnCl_2$ were attempted but the manganese saturated the detector.

ROOL and GOLLD were folded at 1 µM. The samples were diluted 10× prior to taking data. On the stage the samples were further diluted (10 µl buffer:10 µl sample) for a final concentration of 50 nM.

## DLS of RNA nanocages

RNA was folded at 30 ng µl$^{-1}$ using the standard folding protocol. DLS traces were collected using the Prometheus Panta. Two replicates (2 capillaries of 10 µl volume, NanoTemper PR-C002) for each RNA were obtained. DLS data of 10× 5 s acquisitions per capillary with laser power 100% were obtained using PR.PantaControl v.1.8.0. The auto-correlation function was calculated and size distribution was fitted using default parameters in PR.PantaAnalysis v.1.8.0. The resulting size distribution tables and plotting code can be found in the accompanying GitHub repository.

## Cryo-electron microscopy grid preparation

For all samples, the RNA was frozen using a VitroBot Mark IV, using no. 542 filter paper and Quantifoil 1.2/1.3 200 mesh copper grids which were glow discharged for 30 s at 15 mA. GOLLD was folded at 8 µM, using the standard folding conditions except, after the 50 °C incubation, the temperature was lowered to 37 °C at a rate of 0.1 °C s$^{-1}$, held at 37 °C for 2 min, and then reduced to room temperature at a rate of 0.1 °C s$^{-1}$. To increase concentration of GOLLD in the ice, 4 cycles of applying 2 µl of sample and blotting for 3 s were performed before plunging. ROOL was folded at 9.1 µM with the standard folding protocol. The grid was coated with 2 µl of 100 mM NaCl which was blotted for 3 s. Then, 2 µl sample was immediately applied to the grid and blotted for 3 s before plunging into liquid ethane. OLE and raiA motif RNA were frozen with the standard folding protocol at 20 µM and 15 µM respectively; 2 µl of sample was applied to the grid, followed by 3 s blot and plunge into liquid ethane.

## Cryo-electron microscopy data collection

All datasets were collected on Titan Krios G3 microscopes using a 50 µm C2 aperture and 100 µm objective aperture and EPU software (v.3.5). The OLE dataset was collected using a Falcon 4 camera with a 10 eV slit on a Selectris energy filter, while the other datasets were collected using a K3 camera with a 20 eV slit on a Bio Quantum energy filter and EPU software. Additional information on dose, magnification, and data collected for each RNA can be found in Extended Data Table 1.

## Cryo-electron microscopy data processing

Data were processed live using CryoSparc (v.4.5.3)[49] and then further refined, including non-uniform refinement[50]. For OLE and raiA motif per particle motion correction was performed[51]. For all datasets, symmetry was not applied until final refinement stages. For OLE, $C_2$ symmetry was applied. For ROOL and GOLLD, $D_4$ and $D_7$ symmetry were applied, respectively, followed by symmetry expansion of the particles and local refinement for one asymmetric unit. Finally, for GOLLD and ROOL subdomains of one asymmetric unit were locally refined and composite maps, and half-maps were created for one asymmetric unit and then composited to the full symmetry using phenix.combine_focused_maps (v.1.21)[52]. Local resolution was estimated using CryoSparc. See Extended Data Figs. 1–5 for more details on processing pipelines.

## Modelling

Maps were sharpened using phenix.auto_sharpen with half-maps (v.1.21). Initial models for a monomer were obtained from Model-Angelo (Relion-5.0)[53]; because current versions of ModelAngelo cannot be run on a pure RNA structure, EMDB-17659 was added to the corner of the map, the corresponding protein sequence (Protein Data Bank (PDB): 8PHE) was provided, and protein residues were subsequently

deleted from the model. The RNA modelled chains were manually combined tracing the RNA sequences, adding and mutating residues when necessary (in particular, C to U mutations were commonly required). Manual model correction and refinement was accomplished in Coot (version 0.9.8)[54]. Manual refinement of the monomer was performed using Isolde and Coot[54]. Symmetry was applied to the model, from henceforth refinement was done asymmetrically due to limitations in refinement programs. Intermolecular interactions were analysed by hand and corrected using Isolde[55] and Coot[54]. DRRAFTER[56] (Rosetta 3.10 (2020.42)) was used to fill in low resolution areas. For symmetric kissing loops, these models were selected and fit into the map and refined more symmetrically by hand using Isolde[55]. Final refinement was first run through phenix.real_space_refine[57] followed by piecewise corrections using ERRASER2[58] (Rosetta 3.10 (2020.42)), followed by manual refinement in Coot[54] and Isolde[55] when necessary. A protocol was created to enable refinement on the large ROOL and GOLLD complexes. First, the monomer was refined in ~30 sections splitting the model and map prior to using ERRASER2. These were then stitched together and regions encompassing the stitch sites were further refined. Finally, problematic regions of the monomer were refined further. Symmetry was applied to the monomer and the interaction sites were refined in parallel until interactions were sufficiently realistic with only minor clashes. Throughout, split points were manually edited if they caused minimization errors or to include interaction residues. The following ERRASER2 command was used, repeating if not yet converged:

$ERRASER -s $PDB -edensity:mapfile $MAP -edensity::mapreso $RESOLUTION -score:weights stepwise/rna/rna_res_level_energy7beta. wts -set_weights elec_dens_fast 40 cart_bonded 5.0 linear_chainbreak 10.0 chainbreak 10.0 fa_rep 1.5 fa_intra_rep 0.5 rna_torsion 10 suiteness_bonus 5 rna_sugar_close 10 -rmsd_screen 3.0 -mute core.scoring. CartesianBondedEnergy core.scoring.electron_density.xray_scattering -rounds 3 -fasta $FASTA -cryoem_scatterers -rna:erraser:fixed_res $FIXED.

Validation metrics were calculated using Phenix, including phenix.rna_validate[59–61]. ChimeraX (version 1.8)[62] was used to calculate Q-score[61] and for all visuals.

Base pairing and base stacking were identified using Rosetta rna_motif[63]. Kink turns and ribose zippers were identified using DSSR with the "−k-turns" flag (v.1.9.9)[64]. Z-anchors were manually labelled by aligning every 5-nt range of each structure to a representative Z-anchor (4E8Q residues 108−111) and manually inspected each region that had root mean squared deviation (rmsd) < 4 Å. Secondary structure was drawn using RiboDraw with manual manipulation[63]. For visualizing a hypothetical OLE RNA–protein complex, AlphaFold 3 (server version)[47] was used to predict: (1) a OapA dimer with a OapC monomer; and (2) RpsU using the sequences in (Supplementary Table 4). The OapA dimer was fitted into the proposed RNA site manually. The OapC was close to its presumed binding site, but clashed with RNA and therefore its position was manually adjusted. RpsU was also placed manually in its proposed binding site. $C_2$ symmetry was then applied to visualize the full complex.

## Bioinformatic analysis
Bacterial genomes were downloaded from National Center for Biotechnology Information (NCBI) Genome database in February 2024 (https://ftp.ncbi.nlm.nih.gov/genomes/genbank/bacteria/). GenBank records for phage genomes were downloaded in March 2024 (https://millardlab.org/bacteriophage-genomics/phage-genomes-march-2024/). Sequence profiles of GOLLD, ROOL and OLE were downloaded from the Rfam database (ftp.ebi.ac.uk/pub/databases/Rfam/) on March 2024. A custom sequence profile of raiA motif was built using the reported alignment[63]. To retrieve ncRNAs from genome sequences, cmsearch was conducted using sequence profiles with a cutoff value of $10^{-5}$ (Infernal 1.15)[65]. The overall procedure yielded the following numbers of nonredundant ncRNA sequences: 806, GOLLD; 1,596, ROOL; 8,585, OLE; 4,875, raiA motif.

The Infernal software[65] (v.1.1.2) was used to compare candidate RNA structures against Rfam models (cmscan), build and calibrate new covariance models (cmbuild, cmcalibrate) for separate clusters of RNAs, and perform structure-informed homology searches (cmsearch). Comparative analysis and multiple alignments for isolated RNA candidates were conducted using cmalign[65] and MUSCLE (v.5)[66] with pairwise comparisons refined using the OWEN program[67]. Evolutionary history was inferred via the Maximum Likelihood method with different models in MEGA X[68]. Evolutionary analysis of compensatory substitutions in isolated clusters was performed by DecipherSSC[69].

RNAalifold (from ViennaRNA 2.7.0)[70] applied to computationally fold multiple RNA alignments, and Afold/Hybrid[71,72] were used to predict locally folded secondary structures or hybrid duplex elements within clusters. Covariation analysis was performed with R-scape (v.1.2.3)[73], which annotates multiple structural alignments of RNAs using statistically significant covariations (E-value < 0.05) as base-pairing constraints.

## Reporting summary
Further information on research design is available in the Nature Portfolio Reporting Summary linked to this article.

## Data availability
The cryo-EM micrographs and particles, cryo-EM maps and model coordinates have been made available on Electron Microscopy Public Image Archive (EMPIAR), Electron Microscopy Data Bank (EMDB) and Protein Data Bank (PDB), respectively (raiA motif: EMPIAR-12706, EMD-48162 and 9ELY; OLE: EMPIAR-12707, EMD-48163 and 9MCW; ROOL: EMPIAR-12708, EMD-48179 and 9MDS; GOLLD: EMPIAR-12709, EMDB-48214 and 9MEE). Bioanalyzer, DLS and mass photometry data are presented in Supplementary Data 2.

## Code availability
Custom scripts can be found at https://github.com/DasLab/RNA_multimer_2024.

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

**Acknowledgements** The authors thank Stanford Research Computing Center, SLAC Shared Scientific Data Facility, the Stanford-SLAC Cryo-EM Center, the Stanford Chem-H Macromolecular Structure Knowledge Center and the Stanford Biochemistry administrative staff for resources and support that contributed to this research. This work was supported by Stanford Bio-X (Bowes Graduate Student Fellowship to R.C.K.), the National Institute for Health (R35 GM122579 to R.D. and Common Fund Transformative High-Resolution Cryo-Electron Microscopy program U24 GM129541 to W.C.), Howard Hughes Medical Institute (HHMI) (to R.D.), the National Science Foundation (Grant No. 2330652 to R.D. and W.C.), and the G. Harold & Leila Y. Mathers Foundation (MF-2303-04116 to A.G.). S.A.S. and E.V.K. report funding from the Intramural Research Program of the National Institutes of Health of the United States of America (National Library of Medicine). This article is subject to HHMI's Open Access to Publications policy. HHMI lab heads have previously granted a nonexclusive CC BY 4.0 license to the public and a sublicensable license to HHMI in their research articles. Pursuant to those licenses, the author-accepted manuscript of this article can be made freely available under a CC BY 4.0 license immediately upon publication.

**Author contributions** R.C.K., W.C., and R.D. conceptualized and designed the study. R.C.K. and G.N. selected sequences for the study. Y.W. and R.C.K. performed in vitro RNA transcription and collected DLS and mass photometry data. R.C.K. froze cryo-EM grids, collected and processed cryo-EM data, and modelled the cryo-EM maps with advice from W.C. and R.D. G.N. screened cryo-EM grids. H.L. and A.G. generated sequence alignments and S.A.S., R.C.K., R.D. and E.V.K. analysed these sequences for covariation. R.C.K., S.A.S. and R.D. prepared the manuscript with input from all authors.

**Competing interests** The authors declare no competing interests.

**Additional information**
**Correspondence and requests for materials** should be addressed to Wah Chiu or Rhiju Das.

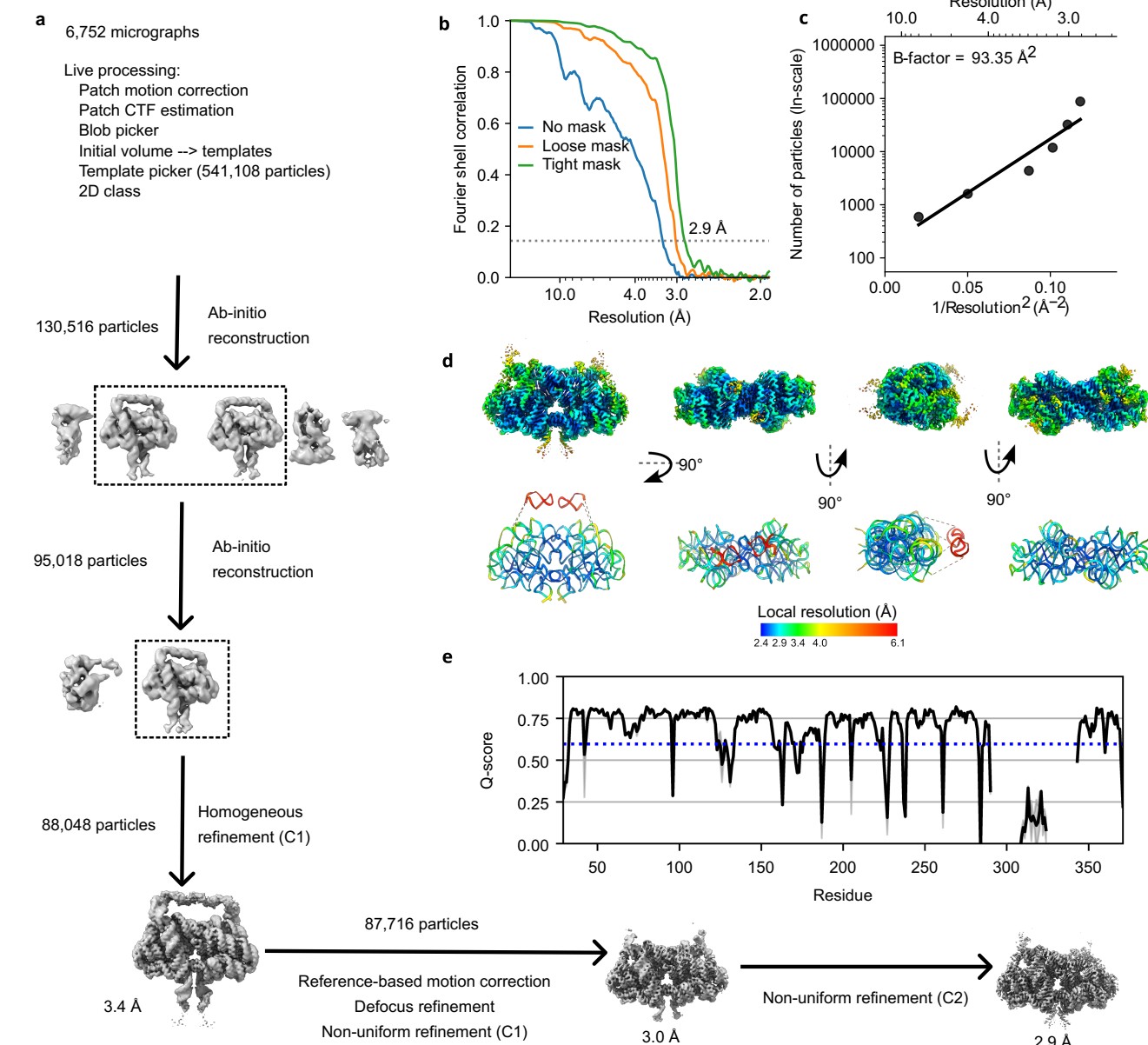

**Extended Data Fig. 1 | Cryo-EM data processing workflow for OLE dimer.**
(**a-e**) OLE resolves into a high resolution dimer, even in the absence of protein. (**a**) Data processing flowchart for the OLE dimer. (**b**) Fourier shell correlation (FSC) plot for final refinement of OLE dimer. (**c**) Plot of particle number against the reciprocal squared resolution for OLE dimer. The B-factor was calculated as twice the linearly fitted slope[74]. (**d**) Local resolution of the OLE dimer on the cryo-EM map (top) and the molecular model (bottom). (**e**) Resolvability of the built model of the OLE dimer as measured by Q-score. The black line is the mean across all chains, with the maximum and minimum values depicted in light grey (N = 2 chains). The expected Q-score at this resolution[72] is labeled with a blue dotted line.

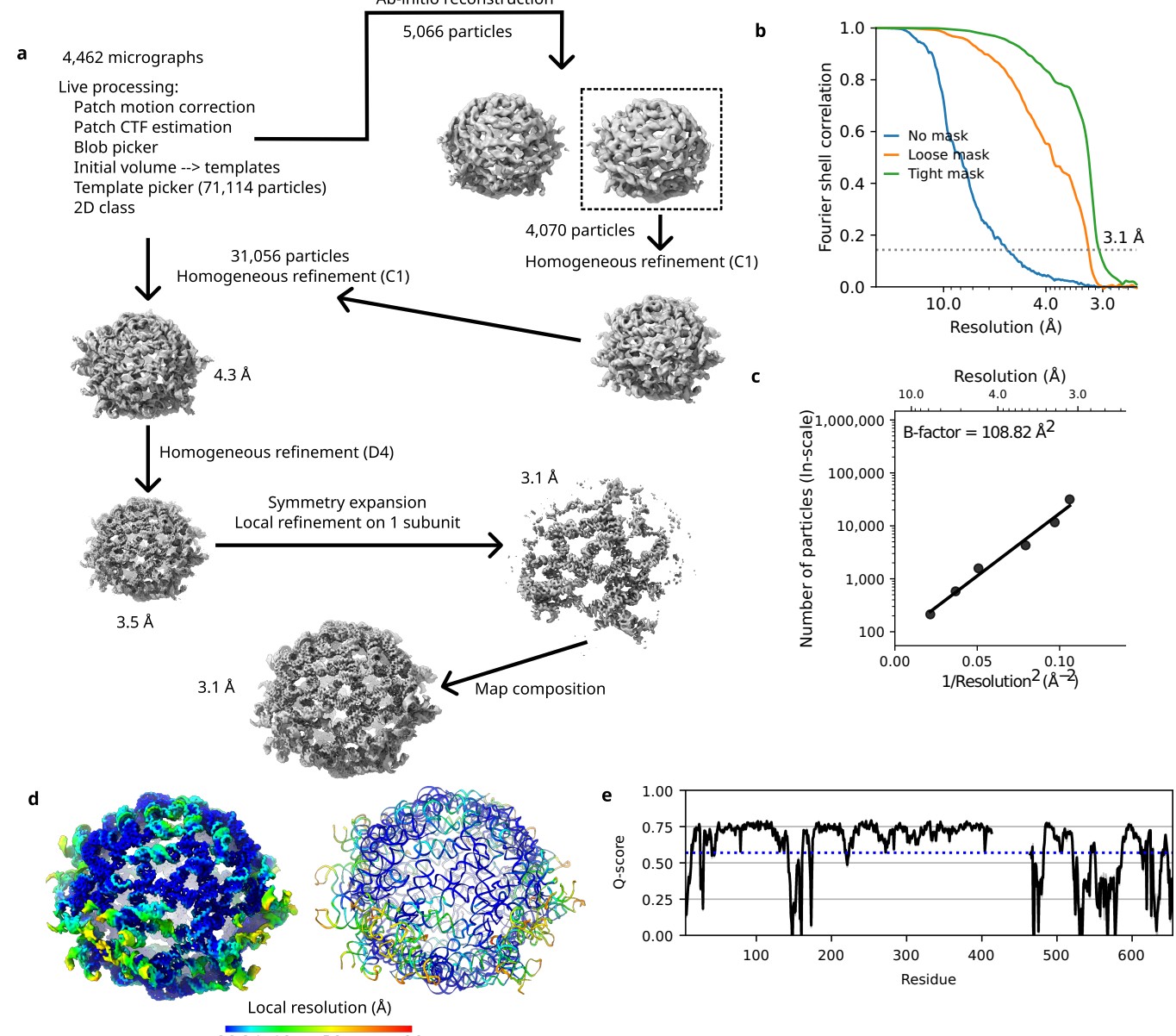

**Extended Data Fig. 2 | Cryo-EM data processing workflow for ROOL nanocage complex.** (**a**) Data processing flowchart. (**b**) Fourier shell correlation (FSC) plots of the single subunit local refinement. (**c**) Plot of particle number against the reciprocal squared resolution for the single subunit local refinement. The B-factor was calculated as twice the linearly fitted slope[74]. (**d**) Local resolution on the cryo-EM map (right) and the molecular model (left). (**e**) Resolvability of the built model as measured by Q-score. The black line is the mean across all chains, with the maximum and minimum values depicted in light grey (N = 8 chains). The expected Q-score at this resolution[72] is labeled with a blue dotted line.

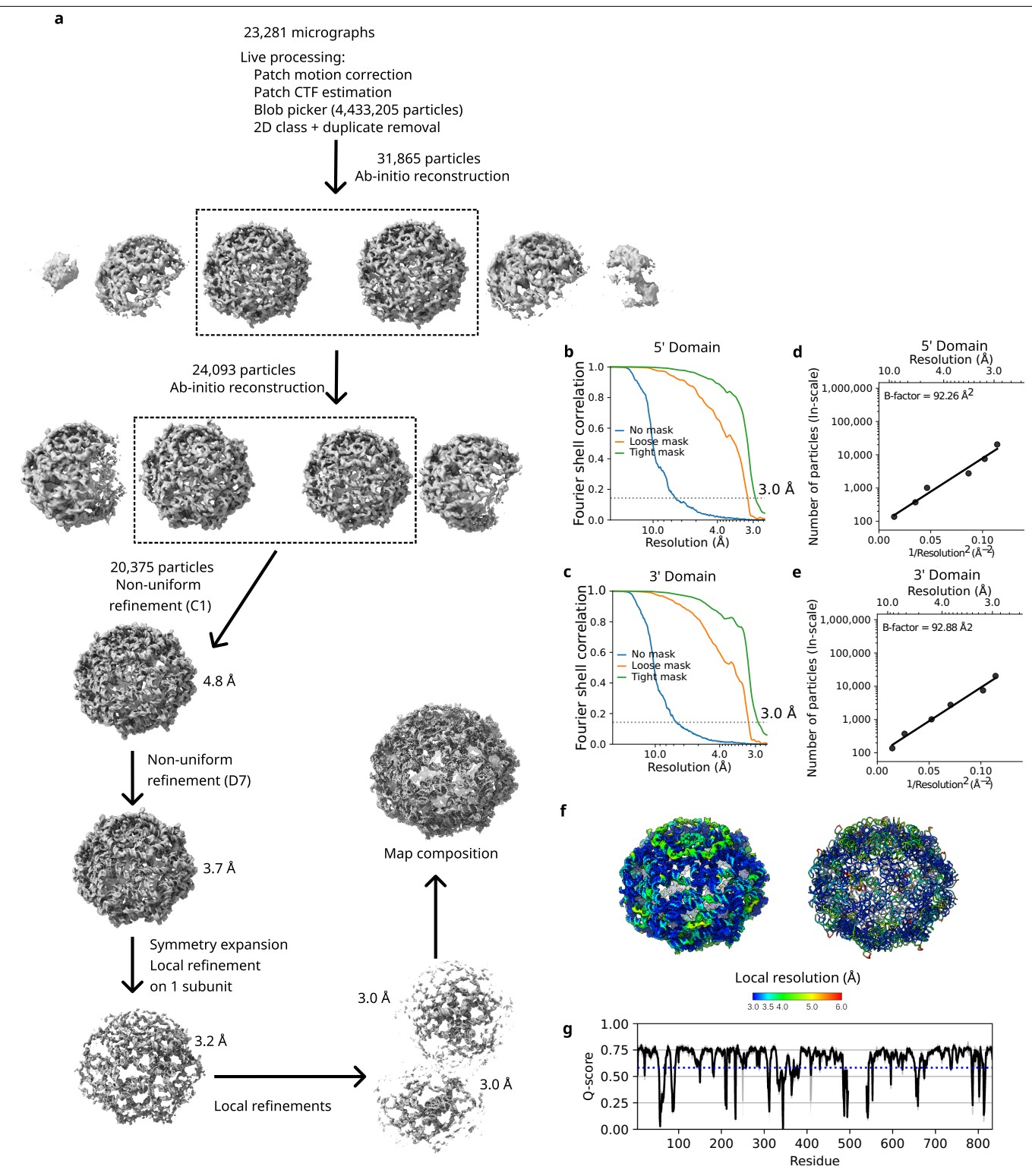

**a**

23,281 micrographs

Live processing:
　Patch motion correction
　Patch CTF estimation
　Blob picker (4,433,205 particles)
　2D class + duplicate removal

31,865 particles
Ab-initio reconstruction

24,093 particles
Ab-initio reconstruction

20,375 particles
Non-uniform
refinement (C1)

4.8 Å

Non-uniform
refinement (D7)

3.7 Å

Symmetry expansion
Local refinement
on 1 subunit

3.2 Å

Local refinements

Map composition

3.0 Å

3.0 Å

**b** 5' Domain

Fourier shell correlation

No mask
Loose mask
Tight mask

3.0 Å

Resolution (Å)

**c** 3' Domain

Fourier shell correlation

No mask
Loose mask
Tight mask

3.0 Å

Resolution (Å)

**d** 5' Domain
Resolution (Å)

Number of particles (In-scale)

B-factor = 92.26 Å²

1/Resolution² (Å⁻²)

**e** 3' Domain
Resolution (Å)

Number of particles (In-scale)

B-factor = 92.88 Å2

1/Resolution² (Å⁻²)

**f**

Local resolution (Å)

3.0 3.5 4.0　5.0　6.0

**g**

Q-score

Residue

**Extended Data Fig. 3 | Cryo-EM data processing workflow for GOLLD nanocage complex.** (**a**) Data processing flowchart. (**b**-**c**) Fourier shell correlation (FSC) plots for the local refinement of the 5′ and 3′ domains respectively. (**d**-**e**) Plots of particle number against the reciprocal squared resolution for the local refinement of the 5′ and 3′ domains respectively. The B-factor was calculated as twice the linearly fitted slope[74]. (**f**) Local

resolution on the cryo-EM map (left) and the molecular model (right). (**g**) Resolvability of the built model as measured by Q-score. The black line is the mean across all chains, with the maximum and minimum values depicted in light grey (N = 14 chains). The expected Q-score at this resolution[72] is labeled with a blue dotted line.

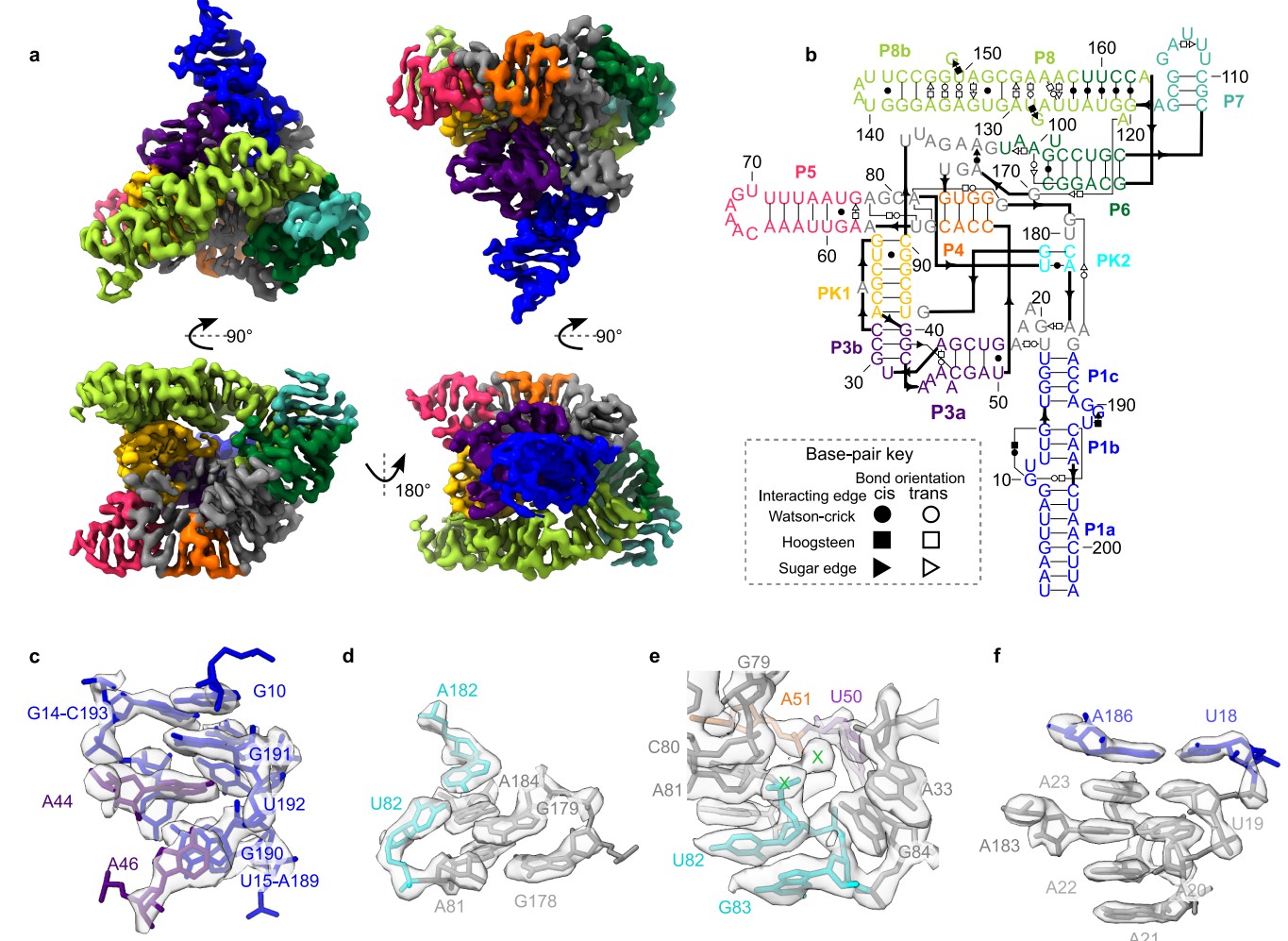

**Extended Data Fig. 4 | Tertiary structure of *raiA* RNA motif.** (**a**) Global view of tertiary structure of *raiA* motif and 2.9 Å cryo-EM map coloured by as labeled in the secondary structure, (**b**). (**c-f**) Select tertiary interactions. Description can be found in Supplemental Text 1. The sharpened cryo-EM map is displayed at the following contours (**a**): 8 σ, (**c,e,f**): 16 σ, (**d**): 20 σ.

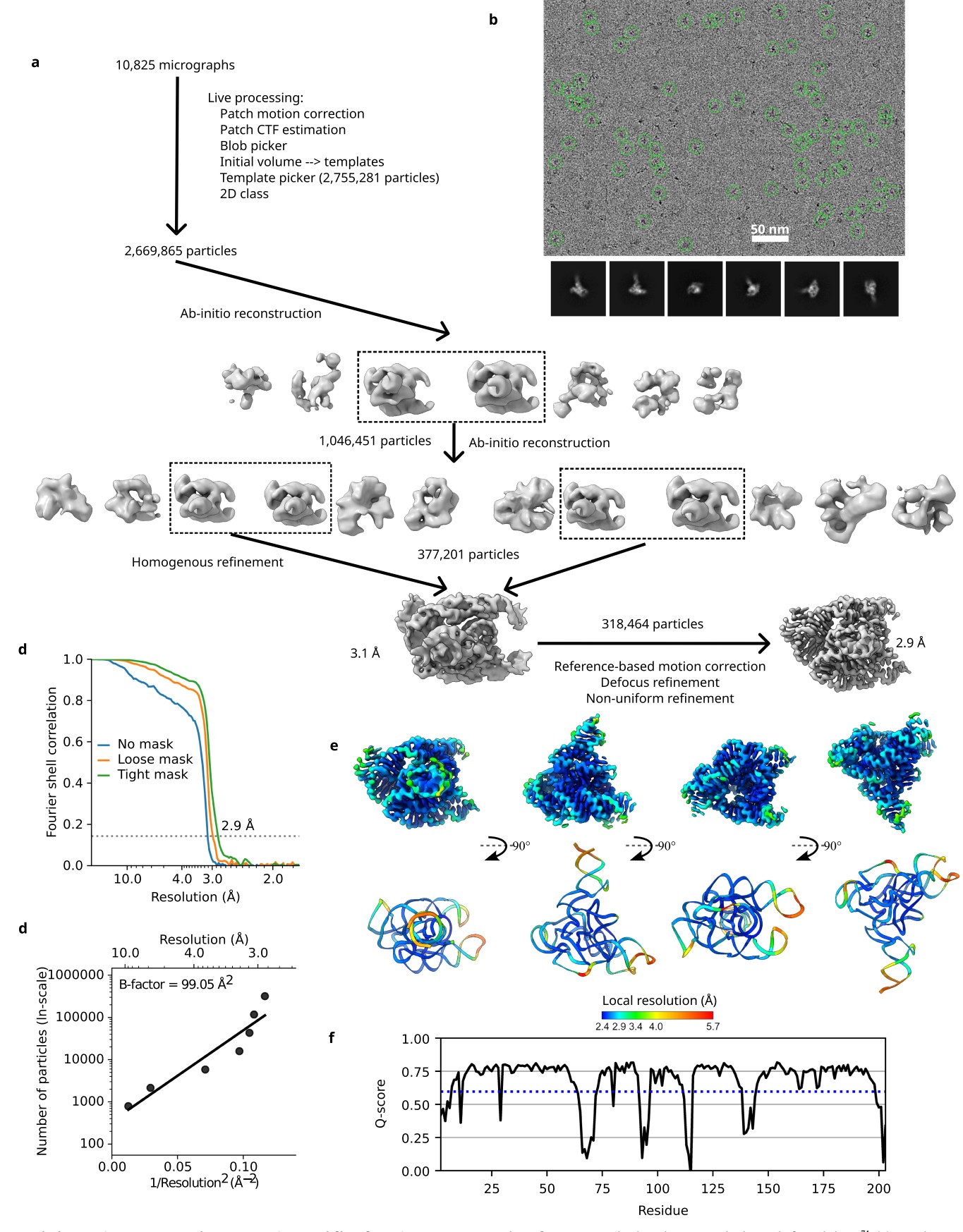

**Extended Data Fig. 5 | Cryo-EM data processing workflow for *raiA* motif.** (**a**) Data processing flowchart. (**b**) Representative micrograph (10,825 micrographs total) and 2D class averages. (**c**) Fourier shell correlation (FSC) plot. (**d**) Plot of particle number against the reciprocal squared resolution. The B-factor was calculated as twice the linearly fitted slope[74]. (**e**) Local resolution on the cryo-EM map (top) and the molecular model (bottom). (**f**) Resolvability of the built model as measured by Q-score. The expected Q-score of a RNA model at this resolution is labeled with a blue dotted line.

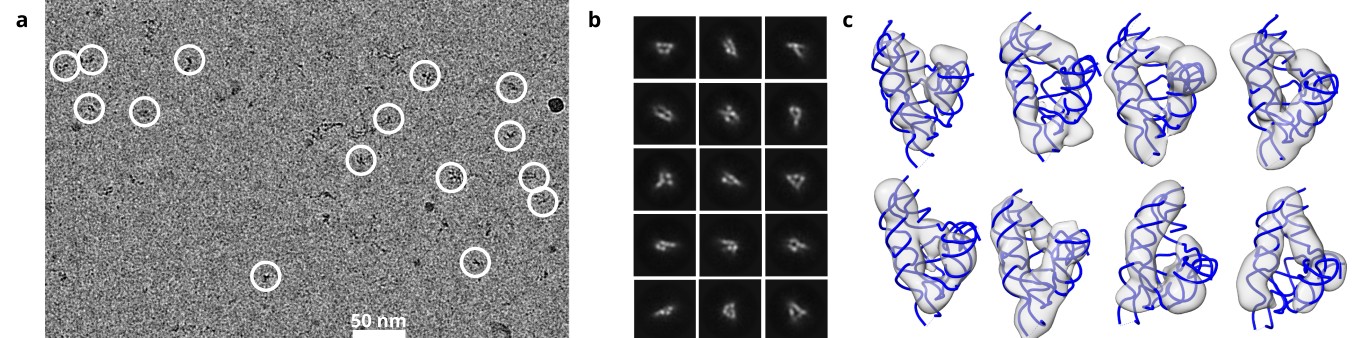

**Extended Data Fig. 6 | Cryo-EM data of HEARO RNA without protein shows disorder.** HEARO did not resolve into a high resolution structure, despite similar amount and quality of data as OLE-dimer. (**a**) The representative micrograph (8,294 micrographs total) of HEARO shows clear particles. (**b**) Select 2D class averages show that HEARO is forming RNA helices, but they have diverse orientations and are blurred, suggesting high flexibility. (**c**) 3D reconstructions of HEARO, overlaid with the known structure of this RNA in the OMEGA nickase complex bound to protein IsrB (PDB: 8DMB[41]), show RNA of a similar fold to the complexed RNA. Multiple conformations are reconstructed, but with poorly resolved features, suggesting that HEARO may not form an atomically ordered structure when not in complex with its partner proteins.

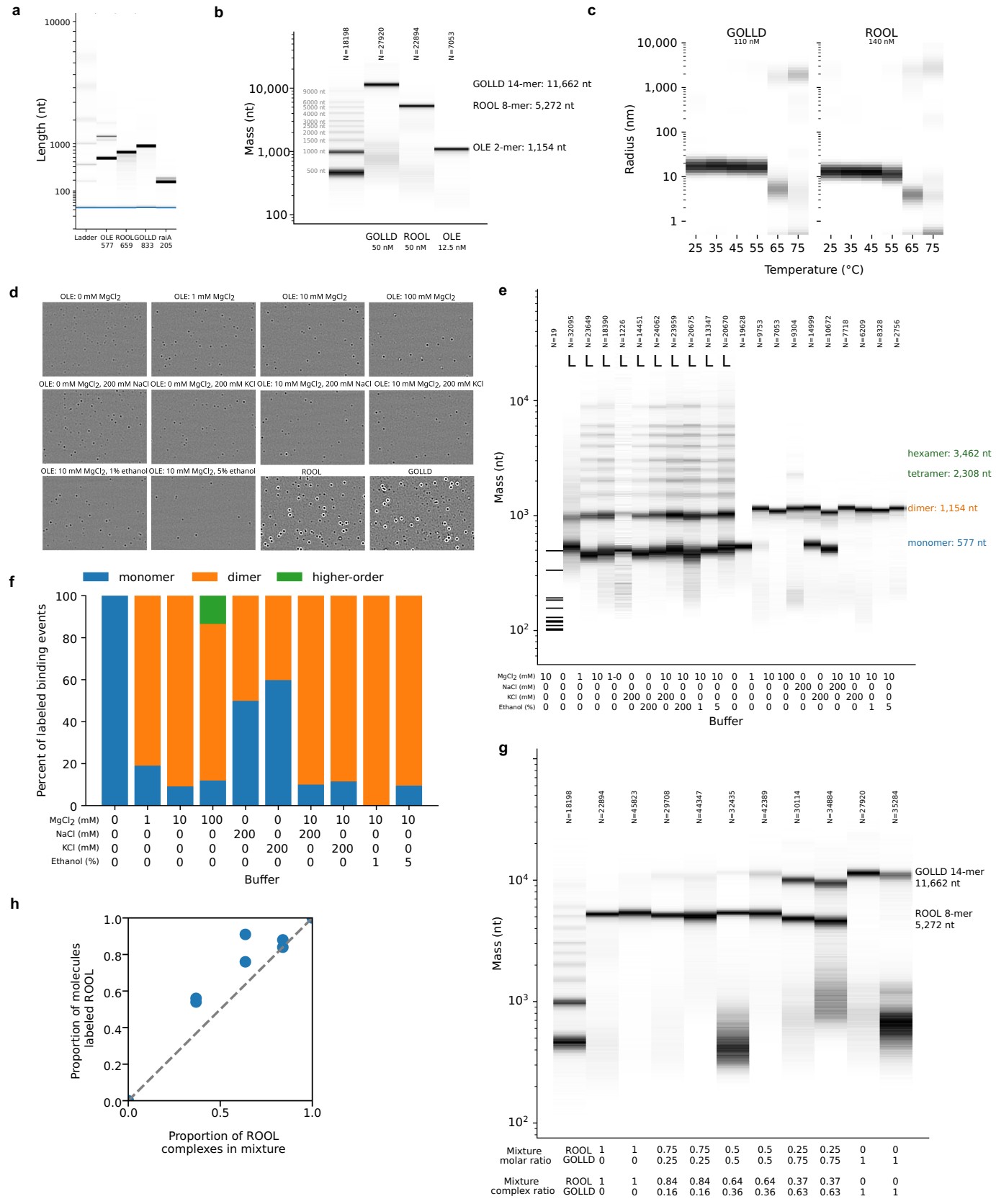

**Extended Data Fig. 7** | See next page for caption.

**Extended Data Fig. 7 | Evidence of multimer formation of GOLLD, ROOL, and OLE in biologically relevant concentrations.** (**a**) Agilent Bioanalyzer traces demonstrate the purity of the samples. The second peak for OLE is a common artifact of poor denaturation of sample in Bioanalyzer traces. The pure monomeric reading in mass photometry, (**b**), shows that this peak is likely not a covalently linked dimer. (**b**) Mass of GOLLD, ROOL, and OLE complexes as obtained from mass photometry at 50 nM, 50 nM, and 12.5 nM respectively. The data is a histogram of particle count density, normalized per sample, where dark is many counts, white is none. Total particle counts are shown above the graph. (**c**) Hydrodynamic radius of GOLLD and ROOL complexes as derived from dynamic light scattering at 110 nM and 140 nM respectively. The data are plotted as relative population density, normalized by density per sample, with dark representing highly populated radius values. The temperature of the sample was raised from 25 °C to 75 °C and dynamic light scattering traces were obtained every 10 °C, showing complex melting into monomers at 65 °C and aggregation at high temperatures. (**d**) Representative ratiometric image for all mass photometry data (1 frame from a 60 s collection at 331 Hz). (**e**) Mass photometry data of OLE in different buffer conditions demonstrates OLE can dimerize at low RNA concentration, low magnesium concentration, and in the absence of magnesium with sufficient monovalent cations. (**f**) The mass photometric data is summarized by counting the amount of hits in the monomer, dimer, and high stoichiometry peaks. The absolute ratio of monomer:dimer is accurate as assessed in (**g-h**). (**g**) Mass photometry traces of mixtures of ROOL and GOLLD, ratiometric image examples can be found in. (**h**) Summary of the mixture results, with the known complex ratio plotted against the ratio reported by mass photometry. There is agreement, but with slight bias towards higher counts for the smaller species, ROOL, opposite of the previously observed trend[75].

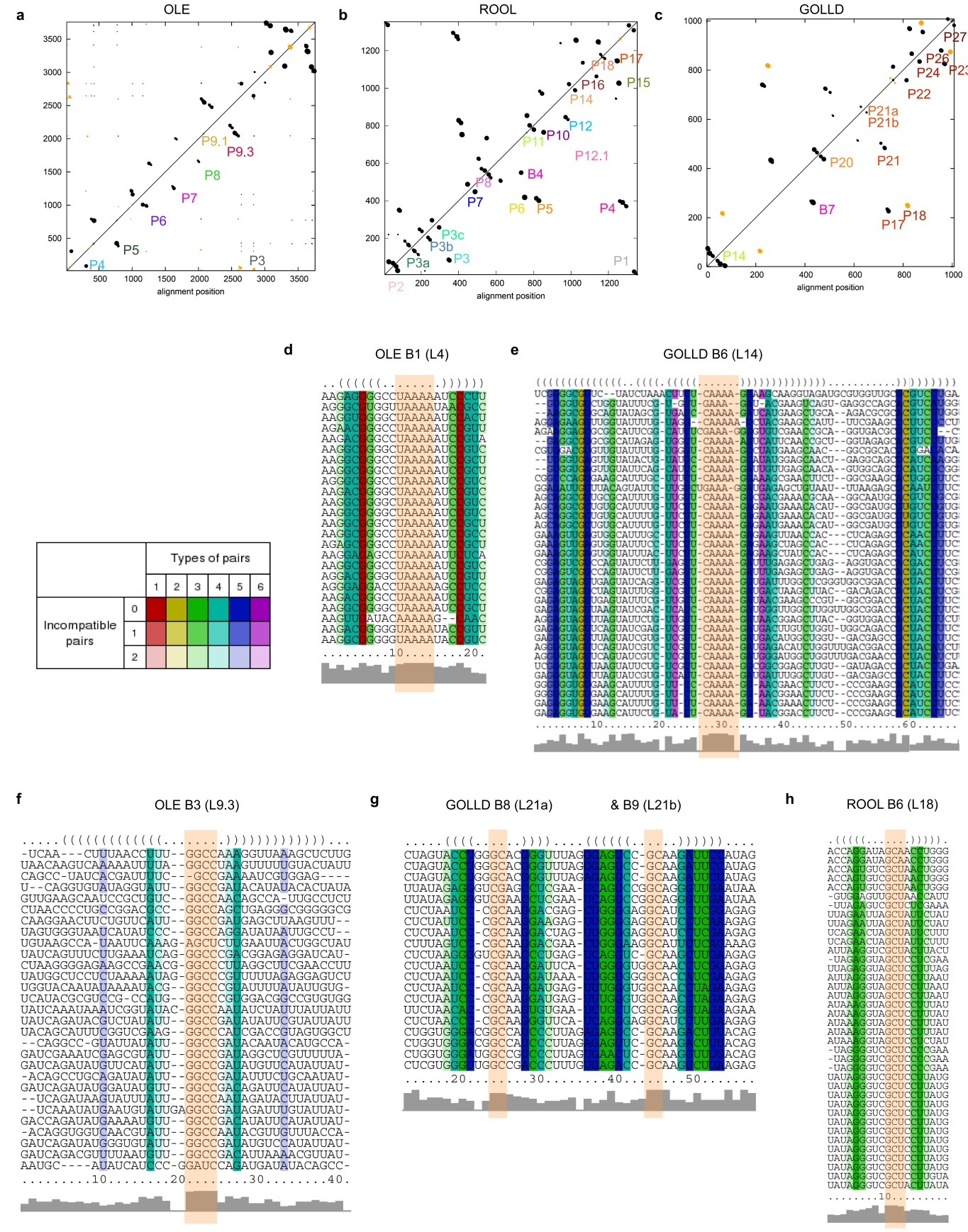

**Extended Data Fig. 8** | See next page for caption.

**Extended Data Fig. 8 | Comparative and covariation sequence analysis of homo-oligomer forming RNAs.** (**a-c**) Distributions of covariation scores in multiple sequence alignments of (**a**) OLE, (**b**), ROOL, and (**c**) GOLLD sequences with select stems labeled. Dot size is proportional to the covariation score. In blue the consensus base pairs are depicted; in green, the consensus base pairs that show significant covariation are shown; in orange, other pairs that have significant covariation were depicted, they are not part of the consensus secondary structure but are compatible with it; in black, other significant pairs are depicted. Positions are relative to the original input alignment (before any gapped column is removed). (**d-h**) Examples of multiple alignments and profiles of sequence identity of selected stable hairpins with highly conserved loops which are involved in the intermolecular interactions are shown. Nucleotides involved in intermolecular interactions are labeled as in main Figs. 1, 2 and 3 for the RNAs OLE (B1, B3), ROOL (B6), and GOLLD (B6, B8) respectively, and highlighted with an orange box. A coloring scheme for highlighting the mutational pattern with respect to the secondary structure (folding) was used and can be found next to (**d**). If one predicted base-pair is formed by several different combinations of nucleotides, consistent or compensatory mutations have taken place. This is indicated by different colors. Pale colors indicate that a base pair cannot be formed in some sequences of the alignment. The sequence variants for the examples were selected from the closest branches of the evolutionary trees built based on the multiple sequence alignments used for the covariation analysis.

**Extended Data Table 1 | Cryo-EM experiments on four large non-coding RNAs**

| | OLE<br>EMPIAR-12707<br>EMDB-48163<br>PDB-9MCW | ROOL<br>EMPIAR-12708<br>EMDB-48179<br>PDB-9MDS | GOLLD<br>EMPIAR-12709<br>EMDB-48214<br>PDB-9MEE | *raiA* motif<br>EMPIAR-12706<br>EMDB-48162<br>PDB-9ELY |
|---|---|---|---|---|
| **Sample preparation** | | | | |
| RNA length (nt) | 577 | 659 | 833 | 205 |
| Molecular weight (kDa) | 187 | 213 | 269 | 67 |
| Stoichiometry | 2 | 8 | 14 | 1 |
| Complex molecular weight (kDa) | 373 | 1,702 | 3,768 | 67 |
| Concentration (µM) | 15 | 9.1 | 8.0 | 20 |
| Concentration (mg/mL) | 2.8 | 1.9 | 2.1 | 1.3 |
| **Image acquisition** | | | | |
| Nominal magnification | 75,000 | 105,000 | 105,000 | 105,000 |
| Voltage (kV) | 300 | 300 | 300 | 300 |
| Electron exposure (e⁻/Å²) | 50 | 50 | 50 | 60 |
| Defocus range (µm) | -1.0 to -2.6 | -0.6 to -2.0 | -1.0 to -2.4 | -0.6 to -2.0 |
| Pixel size (Å) | 0.954 | 0.86 | 0.86 | 0.86 |
| Dose per frame (e⁻/Å²/frame) | 1.25 | 1.25 | 1.25 | 1.5 |
| Exposure time (s) | 4.74 | 1.42 | 2.00 | 1.65 |
| Micrographs acquired | 6,752 | 4,462 | 23,281 | 10,825 |
| **Data processing** | | | | |
| Symmetry imposed | C2 | D4 | D7 | C1 |
| Particle box size* (pix) (extraction size / fourier cropped size) | 360 / 240 | 600 / 400 | 800 / 500 | 320 / 320 |
| Initial particles images | 542,670 | 71,114 | 147,754 | 2,755,281 |
| Final particle images | 87,716 | 31,656 | 20,357 | 318,464 |
| Map resolution (Å) FSC threshold 0.143 | 2.9 | 3.1 | 3.0 | 2.9 |
| Map resolution range (Å) | 2.4-6.1 | 2.9-6.6 | 3.0-6.0 | 2.4-5.7 |
| B-factor (Å²) | 93 | 109 | 93 | 99 |
| **Modeling** | | | | |
| Non-hydrogen atoms | 13,206 | 102,240 | 235,872 | 4,318 |
| RNA nucleotides | 616 | 4,784 | 11,032 | 202 |
| Average Q-score | 0.66 | 0.62 | 0.66 | 0.68 |
| Model resolution (Å) FSC threshold 0.5 | 3.04 | 3.38 | 3.18 | 3.11 |
| Molprobity score | 1.73 | 1.69 | 1.80 | 1.71 |
| R.m.s.d. bond lengths (Å) | 0.015 | 0.003 | 0.003 | 0.003 |
| R.m.s.d. bond angles (°) | 1.583 | 0.620 | 0.620 | 0.703 |
| RNA sugar pucker outliers | 9 (1%) | 0 (0%) | 0 (0%) | 0 (0%) |
| RNA suite outliers | 41 (7%) | 24 (4%) | 654 (6%) | 16 (8%) |
| RNA suiteness | 0.684 | 0.804 | 0.787 | 0.735 |
| Clashscore | 0.20 | 0.10 | 0.42 | 0.15 |

*Additional box size used during processing.

Rhiju Das

# Reporting Summary

## Statistics

For all statistical analyses, confirm that the following items are present in the figure legend, table legend, main text, or Methods section.

| n/a | Confirmed | |
|---|---|---|
| ☐ | ☒ | The exact sample size (*n*) for each experimental group/condition, given as a discrete number and unit of measurement |
| ☒ | ☐ | A statement on whether measurements were taken from distinct samples or whether the same sample was measured repeatedly |
| ☒ | ☐ | The statistical test(s) used AND whether they are one- or two-sided <br> *Only common tests should be described solely by name; describe more complex techniques in the Methods section.* |
| ☒ | ☐ | A description of all covariates tested |
| ☒ | ☐ | A description of any assumptions or corrections, such as tests of normality and adjustment for multiple comparisons |
| ☒ | ☐ | A full description of the statistical parameters including central tendency (e.g. means) or other basic estimates (e.g. regression coefficient) AND variation (e.g. standard deviation) or associated estimates of uncertainty (e.g. confidence intervals) |
| ☒ | ☐ | For null hypothesis testing, the test statistic (e.g. *F*, *t*, *r*) with confidence intervals, effect sizes, degrees of freedom and *P* value noted <br> *Give P values as exact values whenever suitable.* |
| ☒ | ☐ | For Bayesian analysis, information on the choice of priors and Markov chain Monte Carlo settings |
| ☒ | ☐ | For hierarchical and complex designs, identification of the appropriate level for tests and full reporting of outcomes |
| ☒ | ☐ | Estimates of effect sizes (e.g. Cohen's *d*, Pearson's *r*), indicating how they were calculated |

*Our web collection on statistics for biologists contains articles on many of the points above.*

## Software and code

Policy information about availability of computer code

| Data collection | AcquireMP version 2024-R1.1, PR.PantaControl v1.8.0, Bioanlyzer 2100 Expert B.02.11.SI824, EPU 3.5 |
|---|---|
| Data analysis | DiscoverMP version 2024-R1, PR.PantaAnalysis v1.8.0, CryoSparc v4.5.3, phenix 1.21, ModelAngelo (from relion v5), Infernal 1.1.2 and 1.1.5, R-scape 1.2.3, ViennaRNA 2.7.0, Muscle version 5 with OWEN program and MEGA X, Rosetta 3.10 (2020.42), Coot 0.9.8, ChimeraX with ISOLDE and Q-score v1.8, AlphaFold3 server version, DSSR 1.9.9 |

For manuscripts utilizing custom algorithms or software that are central to the research but not yet described in published literature, software must be made available to editors and reviewers. We strongly encourage code deposition in a community repository (e.g. GitHub). See the Nature Portfolio guidelines for submitting code & software for further information.

## Data

Policy information about availability of data

All manuscripts must include a data availability statement. This statement should provide the following information, where applicable:
- Accession codes, unique identifiers, or web links for publicly available datasets
- A description of any restrictions on data availability
- For clinical datasets or third party data, please ensure that the statement adheres to our policy

The cryo-EM micrographs and particles, cryo-EM maps, and model coordinates are made available on EMPIAR, EMDB, and PDB, respectively (raiA motif: EMPIAR-12706, EMDB-48162 and PDB-9ELY; OLE: EMPIAR-12707, EMDB-48163 and PDB-9MCW; ROOL: EMPIAR-12708, EMDB-48179 and PDB-9MDS; GOLLD:

EMPIAR-12709, EMDB-48214 and PDB-9MEE). Bioanalyzer, dynamic light scattering, and mass photometry data can be found in Source Data for Extended Data Figure 7.

# Research involving human participants, their data, or biological material

Policy information about studies with [human participants or human data](). See also policy information about [sex, gender (identity/presentation), and sexual orientation]() and [race, ethnicity and racism]().

| | |
|---|---|
| Reporting on sex and gender | No research involving human participants, their data, or biological material |
| Reporting on race, ethnicity, or other socially relevant groupings | No research involving human participants, their data, or biological material |
| Population characteristics | No research involving human participants, their data, or biological material |
| Recruitment | No research involving human participants, their data, or biological material |
| Ethics oversight | No research involving human participants, their data, or biological material |

Note that full information on the approval of the study protocol must also be provided in the manuscript.

# Field-specific reporting

Please select the one below that is the best fit for your research. If you are not sure, read the appropriate sections before making your selection.

☒ Life sciences ☐ Behavioural & social sciences ☐ Ecological, evolutionary & environmental sciences

For a reference copy of the document with all sections, see [nature.com/documents/nr-reporting-summary-flat.pdf]()

# Life sciences study design

All studies must disclose on these points even when the disclosure is negative.

| | |
|---|---|
| Sample size | Sampling particle images into half maps where the sample size is simply half the total sample: this sampling into two halves is the gold standard method of sampling in cryo-EM so it was chosen such that the resolution we reported is comparable across the field. For dynamic light scattering 10 5 second acquisitions were sampled. This was chosen as the most extensive setting that can be chosen on the Prometheus Panta. Additionally, we saw reproducible results between two replicates. For mass photometry we sampled images at 330 Hz for 60 second totaling 19800. This time was chosen because when the variance of the mass peaks was monitored over time, after this 1 minute the variance no longer decreased significantly, indicating further sampling would not improve the precision of results and thus we had sampled sufficiently. |
| Data exclusions | Cryo-EM particles were excluded in a standard data processing pipeline in order to obtain high resolution data. All the raw data is deposited. No other data was excluded. |
| Replication | In cyro-EM the standard for replication is half-map reconstruction, which means the data is flip into two and two reconstruction are created. This is done for every map and the FSC curves display the resolution up to which the data is "reproducible." Additionally to plot B-factor, a measurement of quality of the cryo-EM data, we repeat this procedure with less and less data. The dynamic light scattering data was conducted in duplicate showing reproducibility between the duplicates. The mass photometry data was examined over the time course of collections showing reproducible peaks when you compare early v late frames. |
| Randomization | Particles are randomly allocated to half in the half-map reconstruction. Earlier in the cryo-EM data process, for filtering out bad particles, those refinement jobs are randomly initiated and run, however we manually select the best particle set to continue on with. For our mas photometry and dynamic light scattering experiments, particles all originated from the same tubes and particle were allowed to randomly diffuse through Brownian motion into our collection area for both methods. There is potential biophysical biases in for example what particle stick to the glass for longer times and hence float into our collection area less. For dynamic light scattering, the solution was mixed before split into replicates for dynamic light scattering which is also a random process. |
| Blinding | Blinding was not necessary in this study as no comparative work was conducted that would cause an unblinded individual to be biased. |

# Reporting for specific materials, systems and methods

We require information from authors about some types of materials, experimental systems and methods used in many studies. Here, indicate whether each material, system or method listed is relevant to your study. If you are not sure if a list item applies to your research, read the appropriate section before selecting a response.

## Materials & experimental systems

| n/a | Involved in the study |
|---|---|
| ☒ ☐ | Antibodies |
| ☒ ☐ | Eukaryotic cell lines |
| ☒ ☐ | Palaeontology and archaeology |
| ☒ ☐ | Animals and other organisms |
| ☒ ☐ | Clinical data |
| ☒ ☐ | Dual use research of concern |
| ☒ ☐ | Plants |

## Methods

| n/a | Involved in the study |
|---|---|
| ☒ ☐ | ChIP-seq |
| ☒ ☐ | Flow cytometry |
| ☒ ☐ | MRI-based neuroimaging |

## Plants

| Seed stocks | *Report on the source of all seed stocks or other plant material used. If applicable, state the seed stock centre and catalogue number. If plant specimens were collected from the field, describe the collection location, date and sampling procedures.* |
|---|---|
| Novel plant genotypes | *Describe the methods by which all novel plant genotypes were produced. This includes those generated by transgenic approaches, gene editing, chemical/radiation-based mutagenesis and hybridization. For transgenic lines, describe the transformation method, the number of independent lines analyzed and the generation upon which experiments were performed. For gene-edited lines, describe the editor used, the endogenous sequence targeted for editing, the targeting guide RNA sequence (if applicable) and how the editor was applied.* |
| Authentication | *Describe any authentication procedures for each seed stock used or novel genotype generated. Describe any experiments used to assess the effect of a mutation and, where applicable, how potential secondary effects (e.g. second site T-DNA insertions, mosiacism, off-target gene editing) were examined.* |

