## [Peer Review File · Nature]

Naturally ornate RNA-only complexes revealed by cryo-EM

Corresponding Author: Professor Rhiju Das

This file contains all reviewer reports in order by version, followed by all author rebuttals in order by version. Additionally, referee#2 in version 1 has provided two figures which are attached at the end of the file.

Version 1:

Reviewer comments:

Referee #1

(Remarks to the Author)

In this work, the authors have characterized the assembly of three fascinating and yet mysterious RNAs: Ole, ROOL and GOLLD, all very long RNAs for which structural data has been lacking. The fundamental advance of this paper (in addition to the rigorous experimental approaches and insights into the RNAs themselves) is that the authors have shown these RNAs to assemble into complex quaternary structures. Such a phenomenon has never been observed previously in natural RNAs, thereby providing fundamental new insights into RNA nanostructural assembly, particularly given the rich comparative analysis of these three natural examples. It gives great strength to the paper that three examples (and three different types of oligomeric states) are presented, as one can begin to discern guiding principles that will inspire work for years to come. In the opinion of this reviewer, this paper will be considered a classic over time, laying the groundwork for innovation in materials and life science. In addition to its foundational significance, there are several other positive features of the paper that are worth noting, which are described below. After that, a few suggestions are included for improving the impact and quality of the paper.

Comments on significance:

A. The cryo-EM approaches, while outstanding in rigor and quality, are not the only distinguishing feature of this paper. One of the most important aspects of the work is that the authors have rigorously characterized the solution assembly state of these RNAs using biophysical and hydrodynamic techniques including mass photometry and electrophoretic methods. This is absolutely essential because the resulting complexes have defined symmetry, which is necessarily leveraged during subsequent cryo-EM processing. The information on oligomeric state therefore provides a rigorous orthogonal backstop for any symmetry constraints imposed in later stages of data processing and analysis. Perhaps most important, the authors show that the specific oligomeric complexes analyzed by cryo-EM are stable (and have apparent K_d values) at very low concentrations – far below that used for sample preparation, and probably lower than their concentrations in the cell. This gives the reader great confidence in the findings of this paper.

B. The cryo-EM analysis is outstanding. The map quality is high and this reviewer appreciated the Q-score values and other metrics of quality for each of the structures described here. This is important because it would be very easy to make significant mistakes in the structure determination of such high symmetry complexes using the types of software and pipelines that are employed in the field. Unrelated to this point, but worth noting – this reviewer is grateful for the carefully organized and choreographed movies which go a long way to helping understand features of the RNAs. Preparation of movies is time-consuming, but helpful to the community.

C. Another feature that gives great confidence to the biological significance of the structures is that the key inter-molecular interfaces are composed of conserved nucleotides, and phylogenetic analysis of the quaternary interactions (and tertiary interactions) strongly supports the resulting structures, often explaining discrepancies in previous biochemical work conducted by chemical probing.

D. The oligomerization strategies of the RNAs themselves is fascinating, involving strand and domain swapped regions of the monomers, and revealing new design principles. To this reviewer, the findings underscore the many parallels between RNA and protein assembly and folding, which are often ignored as so many investigators continue to discount RNA molecules as floppy, ultradynamic polymers, despite the accumulated wealth of information on the specificity and stability of RNA folding. These structures and their strategies for assembly will help correct this issue.

E. This represents an example of structural genomics at its best – in which structures guide our understanding of the biological role for key biomolecules. Only through determining these structures can we begin to understand the likely behavior of Ole, ROOL and GOLLD in living systems. Now one can clearly make reasonable hypotheses for the way in which Ole functions on the membrane, and by which the ROOL and GOLLD cages might contribute to subcellular organization and homeostasis. It is now possible to frame the right questions about these molecules.

Some suggestions:

1. Given the resolution of these structures, the authors could have done a better job characterizing the tertiary interactions contained within them. These molecules are more than Watson-crick and A-minor interactions. There is a wealth of other motifs that this reviewer can clearly observe in the data. A little more time should be taken, particularly with ROOL, to tease out everything in there and annotate it, if even for a subsequent paper. Here is why this is important: A. It is vital for downstream prediction methods that we better characterize the architecture and context for as many RNA tertiary interactions as possible. B. It extends RNA nanotechnology beyond what has been possible for DNA nanotechnology, which relies solely on the use of Watson-Crick interactions for programmable interaction networks. In RNA, we can program A-minors, kink-turns, z-anchors and so much more, and these structures show that this is possible, and how one might do it.

2. There is important, directly relevant work on kink-turn nanostructural assemblies from David Lilley's lab that is not cited or described. It is high quality work that represents an early example of the types of assembly strategies described here, particularly because they are non-W-C. Moreover, the Lilley work shows how different types of motifs and the pitch of their appended helices can lead to different types of shapes and objects. While this reviewer might have missed it, it is important to cite and to discuss briefly in the text, this important work. An example: Huang and Lilley, *Nanoscale* 2016, PMID 27506301. There are several subsequent interesting papers from the lab on these structures, and a review.

Anna Marie Pyle

Referee #2

(Remarks to the Author)

Review – Das, RNA-ornate Nature 2025

This is a significant and elegant accomplishment, opening up new kinds of RNA structure and allowing many new avenues of work. It does an outstanding job of fitting and describing the good parts of the individual monomers, while appropriately not saying much about the low-resolution outer parts, although the use of DRRAFTER did as good a job as feasible of finding approximate structures. And it emphasizes the role of linkers, within and between the chains.

One aspect that deserves more attention than it is given here is the disordered linker. This is conserved in position and approximate length between ROOL and GOLLD, and suggests a role in binding cargo. It is understandable that disorder does not seem important to people who have spent such huge time and effort determining the well-ordered cage structure, but it needs the attention of people who care about disorder and who can perhaps determine its role. As a minimum, mention it on page 12 lines 204-5 by saying "... and a completely empty interior (Fig. 3B and Supplemental Movie 3) except for the disordered loop." Feel free to add additional comment. But it must be in the abstract, to be noticed by the right people. Add something like "Both GOLLD and ROOL form distinct RNA-only multimeric nanocages with diameters larger than the ribosome, empty except for a shared disordered loop."

As emphasized throughout the paper, the interactions both within and between monomers are the most important parts. While they are quite clearly identified they contain a significantly higher number of outliers than the largely helical monomers (although a good deal less than the outer parts). Especially critical is ROOL B5, which has an important other contribution from a different helix of the same chain that comes in from the side to fill the two gaps in your diagram (Fig 2, part N), as shown here in the first stereo figure. Perhaps you did not recognize it because you have the critical residue (A228, in all its replicas) fit backwards, so that it does not H-bond with U507, as shown here in the second figure. Note that the phosphate of 228-9 is entirely out of density (in nearly all replicas) and the base of residue 229 is also fit backward (in most replicas). The conformation could be a real outlier, but it is not this one. We tried refitting this section in COOT and flipped both sidechains to H-bond with their partners and fit the density better. But we had bad geometry outliers in the backbone, since we cannot refine it since we have only maps, not structure factors. You should fix the flipped sidechains and refine this contact as well as feasible, and then add it to contact B5, Fig. 2, part N.

I am somewhat surprised that the manual use of ISOLDE and ERRASER did not fix more of these problems. But there is also some evidence of systematic small shifts of the coordinates to be a bit too far apart at the contacts. For instance, in the inner circle contacts B3, Fig 2, part L, half the contacts in the upper tetramer H-bond G272 to C263 but have outliers in the G272 backbone, and half have a good backbone but are not close enough to H-bond. This may have happened at contact B5 also, so you may not be able to get good geometry in the backbone there. It is only around 1.5Å, but the contacts would not hold the subunits together if they were really like this. In future work you should look into this effect, to see if you can explain this strange behavior. This does not always happen when parts of a structure are aligned separately and then combined, and it does not seem to have happened in GOLLD.

For ROOL, please make the letter used for the chains in the coordinates match what is done in the figures and text. This matters, since the chains were determined separately and differ from one another, and they should also not be in a random order. For ROOL, presuming that chain A = chain 1, then chains A, D, C, and R correspond to the top 4 chains 1, 2, 3, and 4, and E, G, H, and F correspond to the bottom 4 chains 1', 2', 3', and 4'. They should be A, B, C, D and E, F, G, H. For GOLLD, it is OK, and for OLE, it is not an issue.

Minor points:

Caption of Fig. 3, end - give the numbers of the disordered linker. And "as labeled in E"

p 13 line 228 - clearer to say "the distance between the pairs of interacting residues is reduced"

In the intro (p4 line 80-84) you say the RNA you showed was not ordered without its protein was HEARO from *Limnospira maxima*, while on p 15 269-70 you say it was the *raiA* motif from *Clostridium acetobutlicum*, and the data is all described as *raiA*. Please sort out this disconnect, and refer to it also as *raiA* in both places so that readers can connect with the data.

p 15 lines 285-290 - Can you make this point more clearly? It took me 3 readings before I understood. I guess what is missing is what gives the apparent covariance at immediately adjacent pairs, and is it just the closest pair or more than one.

p 23 line 470 - "correct" should be "corrected"

Ext. data Fig. 1, 2, 3, part E - says average is black line and range is shaded, but I don't see those markings

Ext. data Fig 6 - third "B" should be "C"

Ext. data Table 2 - switched EMDB and PDB

Jane Richardson

Version 2:

Reviewer comments:

Referee #1

(Remarks to the Author)

The authors have addressed all of my comments

Anna Marie Pyle

Referee #2

(Remarks to the Author)

You have answered all of my comments, and I recommend accepting your paper with no further changes. It is destined to be a classic.

You have gone above and beyond my asks, and done a great deal of hard work, to fix the problems at interfaces. I am immensely gratified that the hard work was rewarded with much better statistics on validation!

Jane Richardson

Referee #2 Review Attachments 1 and 2

Response to reviewers for: 2024-11-23749B

We thank both expert reviewers for their constructive reviews and detailed look at our structures. In the following, we address each of their points through comments in blue made inline, and end by summarizing additional minor changes.

Referee #1 with expertise in cryo-EM, RNA

In this work, the authors have characterized the assembly of three fascinating and yet mysterious RNAs: Ole, ROOL and GOLLD, all very long RNAs for which structural data has been lacking. The fundamental advance of this paper (in addition to the rigorous experimental approaches and insights into the RNAs themselves) is that the authors have shown these RNAs to assemble into complex quaternary structures. Such a phenomenon has never been observed previously in natural RNAs, thereby providing fundamental new insights into RNA nanostructural assembly, particularly given the rich comparative analysis of these three natural examples. It gives great strength to the paper that three examples (and three different types of oligomeric states) are presented, as one can begin to discern guiding principles that will inspire work for years to come. In the opinion of this reviewer, this paper will be considered a classic over time, laying the groundwork for innovation in materials and life science. In addition to its foundational significance, there are several other positive features of the paper that are worth noting, which are described below. After that, a few suggestions are included for improving the impact and quality of the paper.

Comments on significance:

A. The cryo-EM approaches, while outstanding in rigor and quality, are not the only distinguishing feature of this paper. One of the most important aspects of the work is that the authors have rigorously characterized the solution assembly state of these RNAs using biophysical and hydrodynamic techniques including mass photometry and electrophoretic methods. This is absolutely essential because the resulting complexes have defined symmetry, which is necessarily leveraged during subsequent cryo-EM processing. The information on oligomeric state therefore provides a rigorous orthogonal backstop for any symmetry constraints imposed in later stages of data processing and analysis. Perhaps most important, the authors show that the specific oligomeric complexes analyzed by cryo-EM are stable (and have apparent K_d values) at very low concentrations – far below that used for sample preparation, and probably lower than their concentrations in the cell. This gives the reader great confidence in the findings of this paper.

B. The cryo-EM analysis is outstanding. The map quality is high and this reviewer appreciated the Q-score values and other metrics of quality for each of the structures described here. This is important because it would be very easy to make significant mistakes in the structure determination of such high symmetry complexes using the types of software and pipelines that are employed in the field. Unrelated to this point, but worth noting – this reviewer is grateful for the carefully organized and choreographed movies which go a long way to helping understand features of the RNAs. Preparation of movies is time-consuming, but helpful to the community.

C. Another feature that gives great confidence to the biological significance of the structures is that the key inter-molecular interfaces are composed of conserved nucleotides, and phylogenetic analysis of the quaternary interactions (and tertiary interactions) strongly supports the resulting structures, often explaining discrepancies in previous biochemical work conducted by chemical probing.

D. The oligomerization strategies of the RNAs themselves is fascinating, involving strand and domain swapped regions of the monomers, and revealing new design principles. To this reviewer, the findings underscore the many parallels between RNA and protein assembly and folding, which are often ignored as so many investigators continue to discount RNA molecules as floppy, ultradynamic polymers, despite the accumulated wealth of information on the specificity and stability of RNA folding. These structures and their strategies for assembly will help correct this issue.

E. This represents an example of structural genomics at its best – in which structures guide our understanding of the biological role for key biomolecules. Only through determining these structures can we begin to understand the likely behavior of Ole, ROOL and GOLLD in living systems. Now one can clearly make reasonable hypotheses for the way in which Ole functions on the membrane, and by which the ROOL and GOLLD cages might contribute to subcellular organization and homeostasis. It is now possible to frame the right questions about these molecules.

Thank you for this well-written explanation of the significance of this work for the field.

Some suggestions:

1. Given the resolution of these structures, the authors could have done a better job characterizing the tertiary interactions contained within them. These molecules are more than Watson-crick and A-minor interactions. There is a wealth of other motifs that this reviewer can clearly observe in the data. A little more time should be taken, particularly with ROOL, to tease out everything in there and annotate it, if even for a subsequent paper. Here is why this is important: A. It is vital for downstream prediction methods that

we better characterize the architecture and context for as many RNA tertiary interactions as possible. B. It extends RNA nanotechnology beyond what has been possible for DNA nanotechnology, which relies solely on the use of Watson-Crick interactions for programmable interaction networks. In RNA, we can program A-minors, kink-turns, z-anchors and so much more, and these structures show that this is possible, and how one might do it.

We greatly appreciate this suggestion to annotate more classes of motifs to enhance impact of the manuscript for the fields of RNA structure prediction and RNA nanotechnology. We have now used Rosetta's *rna_motif*, DSSR, and manual inspection to comprehensively identify T-loops, U-turns, GA-minor, loop E submotifs, dinucleotide platforms, UA handles, kink turns, ribose zippers, and z-anchors across all the resolved complexes. Based on this analysis, described in Methods, we have added two tables. **Supplemental Table 1** lists the intermolecular motifs identified. **Supplemental Table 2**, for every bridge, lists the intermolecular base-pairs, base-stacks, and motifs.

Additionally to highlight how rich these structure are in RNA motifs and how that could be useful for prediction and design applications, we have added the following sentence in the Discussion:

“These structures and their complex network of RNA structure motifs offer a rich source of data for RNA structure prediction and design efforts.”

2. There is important, directly relevant work on kink-turn nanostructural assemblies from David Lilley's lab that is not cited or described. It is high quality work that represents an early example of the types of assembly strategies described here, particularly because they are non-W-C. Moreover, the Lilley work shows how different types of motifs and the pitch of their appended helices can lead to different types of shapes and objects. While this reviewer might have missed it, it is important to cite and to discuss briefly in the text, this important work. An example: Huang and Lilley, *Nanoscale* 2016, PMID 27506301. There are several subsequent interesting papers from the lab on these structures, and a review.

Our **Supplemental Table 1** now lists all kink-turns identified in his study, using DSSR to identify these kink turns. We now cite the pioneering work of Lilley et al in reference to our main text discussion of kink turns in OLE.

Anna Marie Pyle

Referee #2 with expertise in RNA structure

Review – Das, RNA-ornate Nature 2025

This is a significant and elegant accomplishment, opening up new kinds of RNA structure and allowing many new avenues of work. It does an outstanding job of fitting and describing the good parts of the individual monomers, while appropriately not saying much about the low-resolution outer parts, although the use of DRRAFTER did as good a job as feasible of finding approximate structures. And it emphasizes the role of linkers, within and between the chains.

Comment 2.1

One aspect that deserves more attention than it is given here is the disordered linker. This is conserved in position and approximate length between ROOL and GOLLD, and suggests a role in binding cargo. It is understandable that disorder does not seem important to people who have spent such huge time and effort determining the well-ordered cage structure, but it needs the attention of people who care about disorder and who can perhaps determine its role. As a minimum, mention it on page 12 lines 204-5 by saying "... and a completely empty interior (Fig. 3B and Supplemental Movie 3) except for the disordered loop." Feel free to add additional comment. But it must be in the abstract, to be noticed by the right people. Add something like "Both GOLLD and ROOL form distinct RNA-only multimeric nanocages with diameters larger than the ribosome, empty except for a shared disordered loop."

We thank the reviewer for the push to discuss the disordered region. The disorder and sequence divergence do warrant hesitancy in overstating relevance of the linker but we agree that it is a region that deserves highlighting to draw the attention of scientists well-suited to tackle such a question. To accomplish this, we have added the phrases in the two suggested locations (abstract and results) as well as the discussion.

Comment 2.2

As emphasized throughout the paper, the interactions both within and between monomers are the most important parts. While they are quite clearly identified they contain a significantly higher number of outliers than the largely helical monomers (although a good deal less than the outer parts). Especially critical is ROOL B5, which has an important other contribution from a different helix of the same chain that comes in from the side to fill the two gaps in your diagram (Fig 2, part N), as shown here in the first stereo figure. Perhaps you did not recognize it because you have the critical

residue (A228, in all its replicas) fit backwards, so that it does not H-bond with U507, as shown here in the second figure. Note that the phosphate of 228-9 is entirely out of density (in nearly all replicas) and the base of residue 229 is also fit backward (in most replicas). The conformation could be a real outlier, but it is not this one. We tried refitting this section in COOT and flipped both sidechains to H-bond with their partners and fit the density better. But we had bad geometry outliers in the backbone, since we cannot refine it since we have only maps, not structure factors. You should fix the flipped sidechains and refine this contact as well as feasible, and then add it to contact B5, Fig. 2, part N.

We appreciate the encouragement for further refinement.

ERRASER2 on these large complexes as one chunk was prohibitively slow. We have previously subset the structure into chunks and refined those, which was also computationally very slow and hence we focused on regions with outliers identified by molprobity. We had additionally noted some convergence issues with ERRASER2, which led to bond outliers that needed to be fixed manually. This manual refinement often ruined the suites determined in ERRASER2. Furthermore, we have identified that as opposed to sampling being the rate limited step in the ERRASER2 protocol as it has been for all smaller RNAs, density calculations actually become the rate limited step here.

So we have now created a script to subset our model and density before running ERRASER2. This has enabled us to refine each chunk of ~50 nt in 8 hours. We additionally have scripted an iterative process so the convergence issues are addressed. Finally, this refinement was run on a monomer, symmetry was imposed, and then the intermolecular interactions were refined, manually checking the results; please see below for regions B5 and B3 of ROOL. This has resulted in higher quality models. Figures for ROOL and GOLLD have been updated in accordance with these new models.

Below please see our PDB and MolProbity score for the full ROOL and GOLLD complexes before and after this refinement. Additionally the fit in map was not significantly negatively affected.

ROOL nanocage before

Clashscore, all atoms:	0.91	
Clashscore is the number of serious steric overlaps (> 0.4 Å) per 1000 atoms.		
Probably wrong sugar puckers:	65	1.99%
Bad backbone conformations ^a :	623	13.02%
Bad bonds:	0 / 114512	0.00%
Bad angles:	459 / 178416	0.26%
Chiral volume outliers	0/23920	
Waters with clashes	0/0	0.00%

ROOL nanocage after

Clashscore, all atoms:	0.23	
Clashscore is the number of serious steric overlaps (> 0.4 Å) per 1000 atoms.		
Probably wrong sugar puckers:	0	0.00%
Bad backbone conformations ^a :	193	4.03%
Bad bonds:	0 / 114504	0.00%
Bad angles:	0 / 178406	0.00%
Chiral volume outliers	0/23918	
Waters with clashes	0/0	0.00%

GOLLN nanocage before

Clashscore, all atoms:	1.82	
Clashscore is the number of serious steric overlaps (> 0.4 Å) per 1000 atoms.		
Probably wrong sugar puckers:	71	1.03%
Bad backbone conformations ^a :	1521	13.79%
Bad bonds:	138 / 264166	0.05%
Bad angles:	1072 / 411670	0.26%
Chiral volume outliers	0/55160	
Waters with clashes	0/0	0.00%

GOLLN nanocage after

Clashscore, all atoms:	0.48	
Clashscore is the number of serious steric overlaps (> 0.4 Å) per 1000 atoms.		
Probably wrong sugar puckers:	0	0.00%
Bad backbone conformations ^a :	654	5.93%
Bad bonds:	0 / 204166	0.00%
Bad angles:	0 / 411670	0.00%
Chiral volume outliers	0/55160	
Waters with clashes	0/0	0.00%

ROOL B5

In particular, this updated process has successfully refined residues 228 and 229 and confirmed that after refinement A228 does pair with U507. The secondary structure diagram was also corrected in Figure 2. Below please see an image of this region.

Comment 2.3

I am somewhat surprised that the manual use of ISOLDE and ERRASER did not fix more of these problems. But there is also some evidence of systematic small shifts of the coordinates to be a bit too far apart at the contacts. For instance, in the inner circle contacts B3, Fig 2, part L, half the contacts in the upper tetramer H-bond G272 to C263 but have outliers in the G272 backbone, and half have a good backbone but are not close enough to H-bond. This may have happened at contact B5 also, so you may not be able to get good geometry in the backbone there. It is only around 1.5Å, but the contacts would not hold the subunits together if they were really like this. In future work you should look into this effect, to see if you can explain this strange behavior. This does not always happen when parts of a structure are aligned separately and then combined, and it does not seem to have happened in GOLLD.

This concern has also been addressed with the re-refinement. This time, only one final round of refinement was applied asymmetrically, reducing the likelihood of major asymmetric features.

ROOL B3

For example, below are the distances between G272 O6 and C263 N4 before and after this refinement, where we can see increased consistency in the distances after refinement.

Before: 4.63, 2.90, 3.29, 3.34, 2.94, 3.22, 3.26, 2.86

New: 3.31, 3.37, 3.44, 3.35, 3.37, 3.37, 3.38, 2.44

Please see the interaction now in the image below. Note the refinement did not result in a completely canonical base-pair (G272 is in the syn conformation rather than the canonical anti conformation), but the positioning of the backbones and bases supports this.

Comment 2.4

For ROOL, please make the letter used for the chains in the coordinates match what is done in the figures and text. This matters, since the chains were determined separately and differ from one another, and they should also not be in a random order. For ROOL, presuming that chain A = chain 1, then chains A, D, C, and R correspond to the top 4 chains 1, 2, 3, and 4, and E, G, H, and F correspond to the bottom 4 chains 1', 2', 3', and 4'. They should be A, B, C, D and E, F, G, H. For GOLLD, it is OK, and for OLE, it is not an issue.

The prior chain labels in the coordinate file were indeed confusing; we have renamed the chains in ROOL as suggested, with A,B,C,D and E,F,G,H now corresponding to 1-4 and 1'-4' in the figure.

Minor points:

Caption of Fig. 3, end - give the numbers of the disordered linker. And “as labeled in E” This has been corrected and nucleotide numbers of disordered linkers are now given in Fig. 2 and Fig. 3.

p 13 line 228 - clearer to say “the distance between the pairs of interacting residues is reduced”

We have made this change which we agree is clearer.

In the intro (p4 line 80-84) you say the RNA you showed was not ordered without its protein was HEARO from *Limnospira maxima*, while on p 15 269-70 you say it was the *raiA* motif from *Clostridium acetobutlicum*, and the data is all described as *raiA*. Please sort out this disconnect, and refer to it also as *raiA* in both places so that readers can connect with the data.

The separate statements on *raiA* and HEARO were confusing in our original manuscript. As it turns out, we have collected cryo-EM data on both HEARO (**Extended Data Figure 6**) and *raiA* motif (**Extended Data Figure 4** and **Supplemental Text 1**). We use these two examples to argue complementary points:

The *raiA* motif is a large ncRNA that we solve in a monomeric form to 2.9 Å resolution. This suggests that our experimental conditions do not induce oligomerization in all large ncRNA.

HEARO is a large ncRNA, known to form a stable complex with a protein which has been structurally determined. The RNA without protein is not well resolved indicating flexibility. This contrasts with OLE forms a stable dimer even without proteins. This further suggests that our experimental conditions are not biasing any of the RNA molecules reported here to oligomerize and be ordered, including RNAs that have protein binding domains that are unoccupied in the experiment.

We have updated the revised manuscript to reflect this by moving both examples to the same section:

“First, concomitant with the same set of cryo-EM studies presented above, we resolved a 2.9 Å resolution map of another large RNA molecule, the *raiA* motif from *Clostridium acetobutylicum*^{23,38}, as a pure monomer (**Extended Data Figure 4-r, Supplemental Text 1**), refuting the possibility that any large RNA would form a multimer in our experimental conditions. Independent group also resolved *raiA* as a monomer^{39,40}. Additionally, we characterized the 343-nt HNH Endonuclease-Associated RNA and ORF (HEARO)⁸ from *Limnospira maxima*, which is known to form a defined RNA structure that is involved in DNA nickase activity when bound to the protein *IsrB*²⁵. Unlike OLE 5' region, which is also known to bind proteins, the HEARO RNA was disordered in the absence of the protein (**Extended Data Figure 6**), suggesting multimer formation of protein-binding RNAs is not an artifact of cryo-EM experimental conditions.”

To aid in readability of this section, we have swapped the order of this paragraph with the previous two in the original manuscript.

p 15 lines 285-290 - Can you make this point more clearly? It took me 3 readings before I understood. I guess what is missing is what gives the apparent covariance at immediately adjacent pairs, and is it just the closest pair or more than one.

We have made the following replacement:

“In other bridges, we observed intermolecular symmetric kissing loops which had base-pairs between the same loop from two different chains: nucleotides 315-318 in chain A and B of OLE (B3) and nucleotides 656-657 from chain 1 and 7' of GOLLD (B8). Apparent covariance at immediately adjacent nucleotides in these loop sequences support intermolecular base-pairs because base-pairing of adjacent nucleotides within the same chain is stereochemically precluded (Extended Data Fig. 8F-H).”

p 23 line 470 - “correct” should be “corrected”
This edit has been made.

Ext. data Fig. 1, 2, 3, part E - says average is black line and range is shaded, but I don't see those markings

The shaded region is the light grey marking on these figures (see below). Most of the residues had near equal Q-scores across chains, with deviations only visible in the poorly resolved regions. We have clarified the legend by saying “depicted in light grey.”

Ext. data Fig 6 - third “B” should be “C”
Ext. data Table 2 - switched EMDB and PDB
These 2 edits have been made.

Jane Richardson